# Cyclic Nucleotide Phosphodiesterase Families as Targets to Treat Pulmonary Arterial Hypertension: Beyond PDE5 Inhibitors?

**DOI:** 10.3390/cells14211670

**Published:** 2025-10-25

**Authors:** Liting Wang, Rodolphe Fischmeister, Boris Manoury

**Affiliations:** 1Université Paris-Saclay, Inserm, UMR-S 1180, 91400 Orsay, France; wanglitingfw@163.com (L.W.); rodolphe.fischmeister@inserm.fr (R.F.); 2Department of Cardiology, Guangdong Cardiovascular Institute, Guangdong Provincial People’s Hospital, Guangdong Academy of Medical Sciences, Southern Medical University, No. 106 Zhongshan 2nd Road, Guangzhou 510080, China

**Keywords:** phosphodiesterases, pulmonary arterial hypertension, cAMP, cGMP, pharmacological therapies

## Abstract

Pulmonary arterial hypertension (PAH) is a fatal disease with no cure. Until recently, most specific therapies for PAH had aimed at enhancing cyclic nucleotide (cAMP and cGMP) pathways, taking advantage of the vasorelaxant and antiproliferative properties of these key intracellular messengers. This process can be achieved by inhibiting phosphodiesterases (PDEs), which are intracellular enzymes responsible for cyclic nucleotide degradation. To date, only inhibitors of PDE type 5 (PDE5) have been approved for the treatment of PAH. Because the PDE superfamily comprises 11 families that encompass many variants, substantial experimental investigation has been conducted to assess the relevance of inhibiting other PDE families, aiming to offer therapeutic alternatives. This review synthesizes the main research work conducted on in vivo or ex vivo models, as well as on biological resources from patients. It helps provide evidence for the expression of PDE isoforms in the lung vasculature, as well as the efficacy and limitations of various pharmacological compounds tested for inhibiting pathological processes ongoing in the disease. Perspectives and suggestions for future research orientation are proposed.

## 1. Pulmonary Arterial Hypertension

### 1.1. Introduction

While pulmonary arterial hypertension (PAH) is a rare disease (48–55 cases/million), it is life-threatening and non-curable [1]. In PAH, progressive elevation of pulmonary vascular resistance leads to right heart failure and premature death. Over the last few decades, a few specific therapeutic options have improved survival rates and patient quality of life but have remained insufficient in reversing the progression of the disease. Therefore, research efforts have focused on better understanding the mechanisms of the disease and the diversification of therapeutic targets. Recently, sotatercept, a first-in-class activin signalling inhibitor, has yielded promising results in clinical trials [2,3]. Sotatercept can now be combined with classical PAH therapies [4], and future clinical investigations will aim at refining the strategies for optimal efficacy and safety. Drugs that promote the beneficial effects of cyclic nucleotide signalling in the lung vasculature, namely prostacyclin (PGI_2_) analogues and phosphodiesterase-type 5 inhibitors (PDE5i), remain important pillars of the disease management [4]. In the context of therapeutic research, since modulation of cyclic nucleotides can be achieved by various pharmacological means, it is crucial to investigate further options in order to optimize their potential. In particular, cyclic nucleotide phosphodiesterases (PDEs), as a very large and diverse family of druggable enzymes, may provide further insights for disease management [5]. This review synthesizes the research that sought to extend the field of PDEs as relevant pharmacological targets in PAH, beyond the approved class of PDE5i.

### 1.2. Definition, Classification, and Pathophysiology of PAH

Pulmonary hypertension (PH) is a heterogeneous cardiopulmonary disease, characterized hemodynamically by a mean pulmonary arterial pressure (mPAP) higher than 20 mmHg at rest, standardly measured by right heart catheterization [6]. PH results from a myriad of clinical conditions and thus is divided into five groups, which are: pulmonary arterial hypertension (PAH, group I), PH associated with left heart disease (group II), PH associated with lung diseases and/or hypoxia (group III), PH associated with pulmonary artery obstructions (group IV), and PH with unclear and/or multifactorial mechanisms (group V) [6] (Table 1). PAH is diagnosed by the concomitant presence of mPAP > 20 mmHg, pulmonary arterial wedge pressure ≤ 15 mmHg, and pulmonary vascular resistance > 2 WU at rest, and exclusion of groups III, IV, and V PH [6]. PAH is a primary disorder of the pulmonary vasculature and includes patients with similar pathophysiological, histological, and prognostic features, of variable causes. PAH is further classified into 6 subgroups, according to etiology or clinical manifestations (Table 1).

The pathological manifestations of pulmonary vasculopathy include hypercoagulation, vasoconstriction, remodelling of the pulmonary vessel wall caused by endothelial hyperproliferation or plexiform lesions, hyperplasia of pulmonary arterial smooth muscle cells (PASMCs), and hypertrophy. This is featured by inflammatory cell recruitment and fibrosis in the adventitia, as well as immune dysregulation [7].

### 1.3. Current Therapeutic Options

Prior to 1995, there were no specific treatments for PAH. Pharmacological management mainly used L-type calcium-channel blockers, anticoagulation therapy, digitalis, and diuretics to relieve symptoms of right ventricular failure. Consistently, the prognosis of PAH was very poor, with a 1-, 3-, and 5-year survival of 68%, 48% and 34%, respectively, and the median survival was 2.8 years [8]. The new pharmacological classes that emerged mainly aim to rescue the function of the pulmonary endothelium, which is impaired in PAH.

Endothelial dysfunction is typically characterized by chronically impaired bioavailability of paracrine, vasodilating mediators, such as nitric oxide (NO) and PGI_2_. Meanwhile, expression of some vasoconstrictors, such as endothelin-1 (ET-1), is increased. Treatment strategies were developed to target these specific pathways (reviewed in [9]): first, enhancing PGI_2_ pathway using pharmacological analogues (epoprostenol, treprostinil, and iloprost) or a synthetic agonist of the IP receptor, namely selexipag; second, enhancing NO-3′,5′ cyclic guanosine monophosphate (cGMP) pathway using PDE5i (sildenafil and tadalafil) or soluble guanylyl cyclase (sGC) stimulator (riociguat); third, repressing ET-1 signalling by using receptors antagonists (bosentan, ambrisentan, and macitentan). These pharmacotherapies have largely improved PAH survival [10,11], although the 5-year survival rate for these patients remains less than 60% [12,13]. Lung transplantation or heart–lung transplantation is regarded as the only way to cure this disease. Recently, sotatercept, a biological therapy approved by the FDA in 2024, has been developed to tackle the disease mechanism through a novel pathophysiological angle. This drug aims to impede the TGF-β–activin–nodal branch signalling by binding to ligands of deleterious downstream receptors. This would improve the balance toward the beneficial bone morphogenetic protein (BMP)–growth differentiation factor (GDF) branch signalling and therefore prevent the deleterious remodelling of the pulmonary arterial wall that underlies PAH [14]. The Phase 3 STELLAR clinical trial showed that sotatercept increases exercise capacity (6-min walk distance), improves WHO functional class and NT-proBNP levels, and reduces the risk of clinical worsening events in PAH patients [2]. Sotatercept is now part of the therapeutic arsenal that can be used in PAH [4]. The three classes previously described, administered initially or sequentially as part of a bi- or tri-therapy, are still recommended for managing moderate to severe PAH as shown in clinical guidelines [4,6]. In addition, the management of PAH may benefit from increased clinical experience, refinement of the treatment algorithm, and increased access to pharmaceuticals due to patent expirations and the advent of generic drugs.

## 2. Cyclic Nucleotide Pathways as Therapeutic Targets in PAH

### 2.1. General Roles of Cyclic Nucleotide Pathways in Circulation

Cyclic nucleotides, namely 3′, 5′-cyclic adenosine monophosphate (cAMP) and 3′, 5′-cyclic guanosine monophosphate (cGMP), are ubiquitous second messengers involved in the intracellular signal transduction of specific extracellular stimuli (summarized in Figure 1). In the cardiovascular system, they are key regulators of multiple normal biological processes such as excitation-contraction coupling, blood vessel vasodilation, and natriuresis. In pulmonary circulation, cAMP and cGMP contribute to the highly dilated status of the pulmonary artery (PA) [9,15,16], thus ensuring low pulmonary vascular resistance and PA pressure. Importantly, cyclic nucleotides also interfere with some pathological processes, including tissue remodelling or inflammation. In the vasculature, cAMP- and cGMP-mediated pathways are generally regarded as protective because they mediate vasodilation, exert antiproliferative and pro-apoptotic effects on smooth muscle cells, inhibit platelet activity, and maintain the endothelial barrier [17,18,19,20,21]. Enhancing both cAMP and cGMP pathways results in promoting BMPR2 and Smad1/5 signalling, which are important for maintenance of the pulmonary vascular integrity [22,23]. Because sotatercept also aims to restore these pivotal mechanisms, cyclic nucleotide-based therapies may provide supplemental benefit in targeting the TGF-β axis.

Therefore, consensus in PAH is to favour both pathways in pulmonary circulation as an attempt to reduce the progressive increase in pulmonary vascular resistance (Figure 2).

Stimulation of cGMP may also favour the condition of the right ventricle (RV), by limiting cardiac deleterious remodelling of the myocardium [24,25]. By contrast, stimulating cAMP in the cardiac myocytes is considered deleterious, as it promotes hypertrophy, cell death, and cardiac fibrosis [24]. Importantly, strategies aiming at preserving the cAMP pathway in PAH may be limited and may have to be restricted to the vascular territory.

The biological activity of cyclic nucleotides directly depends on their respective intracellular levels. Thus, therapeutic interventions modulate these pathways by altering the balance between the rates of synthesis or elimination of cAMP and cGMP.

### 2.2. Molecular Determinants of the cAMP and cGMP Pathway

#### 2.2.1. Synthesis of cGMP and cAMP

The modalities of cAMP and cGMP synthesis have been extensively described in dedicated reviews [15,16,17,26,27], and will only be summarized here and in Figure 1.

##### cGMP

Synthesis of cGMP in vascular smooth muscle cells is catalyzed by two classes of enzymes, which convert GTP into cGMP: (i) sGC, a cytosolic heme-containing heterodimer, typically produces cGMP upon binding to nitric oxide (NO); (ii) particulate guanylate cyclases (pGC), which comprise 2 membrane receptors for natriuretic peptides (NPs), namely GC-A and GC-B (also called NPR-A, -B). These membrane proteins produce cGMP by an intracellular catalytic activity. GC-A and GC-B can be stimulated by different NPs, including ANP, BNP or CNP, with various potencies [17,26]. All these pathways were reported to be active in PA [15,16,28]. The cGMP-activated protein kinase (PKG) is one important effector of cGMP.

##### cAMP

Synthesis of cAMP from ATP is achieved by adenylyl cyclases (ACs). There are nine membrane-bound AC isoforms and one soluble (sAC) isoenzyme, each with distinct expression and regulatory profiles [29]. Several isoforms were described in PASMCs, with variations among species [30,31]. ACs are classically stimulated following stimulation of G(α)s protein-coupled receptors by their respective agonists. In vascular cells, such receptors include β-adrenoreceptors, IP receptors (stimulated by PGI_2_), or A2 adenosine receptors, among others [19,32]. In the vasculature, cAMP primarily exerts its effects via the activation of cAMP-activated protein kinase (PKA) and exchange protein activated by cAMP-1 (EPAC1) [19].

#### 2.2.2. Hydrolysis of cGMP and cAMP by PDEs

Intracellular levels of cAMP and cGMP are mainly mitigated by the PDEs. These enzymes hydrolyze cyclic nucleotides into non-cyclic 5′-GMP or 5′-AMP, which are unable to activate cyclic nucleotide effectors. PDEs comprise a superfamily of enzymes classified into 11 families (Table 2), altogether encoded by more than 20 genes [5,33,34,35]. PDE families notably differ by their affinity (Km) to either cyclic nucleotide: some PDEs (types 5, 6, and 9) are cGMP specific, some are cAMP-specific (PDE4, PDE7, PDE8), while others are dual specific (Table 2). PDE families also distinguish themselves through various regulatory mechanisms, including possibilities for post-translational modifications, and feedback regulation by cGMP or cAMP through binding to “GAF” domains (Table 2). Additionally, as cGMP interacts with some PDEs that mitigate cAMP, crosstalk between pathways can be established (see Section 3.2 and Section 3.3).

In addition to PDEs, export of cyclic nucleotides by multidrug resistance-associated protein 4 (MRP4, also known as Abcc4) was demonstrated in arterial smooth muscle cells [36]. This mechanism is of pathophysiological relevance because MRP4 expression has been found to increase in PAH, and the inhibition or deletion of MRP4 has been shown to prevent the development of hypoxia-induced PH in mice [37].

**Table 2 cells-14-01670-t002:** Properties of PDEs and their distribution in the normal pulmonary artery.

PDE	Gene	Km (µM) [38]	Substrate Selectivity and Salient Features	Pulmonary Artery Activity/Expression	Inhibitors (IC_50_, nM)
PDE1	PDE1A	cAMP: 73–120 cGMP: 2.6–5	dual substrate Ca^2+^/calmodulin-regulated	Vessel: (+) [39,40,41,42] VSMC: (+) [43,44,45] (PDE1C highly expressed in proliferative phenotype) [43] EC: ND	8MM-IBMX (NS) vinpocetine: 14,000
	PDE1B	cAMP: 10–24 cGMP: 1.2–5.9			
	PDE1C	cAMP: 0.3–1.2 cGMP: 0.6–2.2			
PDE2	PDE2A	cAMP: 30–50 cGMP: 10–30	dual substrate cGMP-stimulated	Vessel: (+) [39,40,42,46,47] VSMC: (+) [44] EC: (+) [48]	EHNA: 800 Bay 60-7550: 4.7
PDE3	PDE3A PDE3B	cAMP: 0.02–0.15 cGMP: 0.18	dual substrate cGMP-inhibited	Vessel: (++) [39,40,42] VSMC: (+) [44,49] EC: (+) [48]	cilostamide: 25–50 cilostazol: 200 milrinone: 150
PDE4	PDE4A PDE4B PDE4C PDE4D	cAMP: 2.9–10	cAMP-selective	Vessel: (++) [39,40,42] VSMC: (++) [44] EC: (+) [48]	rolipram:1000 cilomilast: 70–120 roflumilast: 0.4–0.6
PDE5	PDE5A	cGMP: 1–6.2	cGMP-binding, cGMP-selective	Vessel: (+++) [39,40,42] VSMC: (+++) [43,49,50] EC: ND	zaprinast: 500–700 sildenafil: 5 vardenafil: 1 tadalafil: 5
PDE6	PDE6A	cGMP: 15–17	cGMP-selective, photoreceptor	not expressed [34]	ND
PDE7	PDE7A PDE7B	cAMP: 0.1–0.2	cAMP-selective	VSMC: (+) (mRNA) [51,52] EC: ND	rolipram-insensitive
PDE8	PDE8A PDE8B	cAMP: 0.04–0.06	cAMP-selective	VSMC: (+) (human mRNA) [52]; (−) (rat mRNA) [51] EC: ND	ND
PDE9	PDE9A	cGMP: 0.17–0.39	cGMP-selective	VSMC: (±) (mRNA) [51,52] EC: ND	Bay 73-6691: 55 PF-04447943: 2.8
PDE10	PDE10A	cAMP: 0.26 cGMP: 7.2	cAMP-inhibited, dual substrate	VSMC: (+) (mRNA) [52]; (−) (rat mRNA) [51]; (immunoreactivity) [53] EC: ND	papaverine: 36
PDE11	PDE11A	cAMP: 1.04–5.7 cGMP: 0.52–4.2	dual substrate	VSMC: (+) (mRNA) [52] EC: ND	ND

VSMC: vascular smooth muscle cell; EC: endothelial cell; NS: non-selective; ND: not documented; (−): no expression detected; (±): no change in expression; (+), (++), to (+++): moderate, fair or high expression.

PDEs have been considered for a long time as druggable targets, and some PDE inhibitors are used in various therapeutic areas (Table 3; reviewed in [5,35]).

### 2.3. General Modulation of cAMP and cGMP Pathways in PAH

#### 2.3.1. Alterations of cAMP and cGMP Levels in PAH

Under normal conditions, cGMP concentration is around 2 times higher than cAMP in rat PA [54], although the cAMP levels are reported around 5 times higher than cGMP in systemic artery [19]. PAs isolated from hypoxic rats showed decreased cAMP in large vessels [54]. Likewise, stimulation of PASMCs from PAH patients with forskolin, a direct activator of AC, was reduced [52]. Evidence suggests that expression or activity of some AC isoforms can be depressed by various pathological stimuli (cytokines, hypoxia) [31,55].

Similar to the changes observed in cAMP, the level of cGMP content in PA from hypoxic rats was decreased [54]. By contrast, cGMP in lung perfusate was found to be higher in a PH model [56]. In the latter study, because cGMP-PDE activity was unchanged, the authors proposed that production of cGMP in the lungs should be enhanced following chronic hypoxia (CHx) [56]. One cannot rule out the possibility that high circulating levels of NPs resulting from cardiac overload lead to increased cGMP release in vivo [56]. Expressions of some sGC subunits were found to increase in PAH donors and models, although the identity of modulated genes or proteins varies among detection methods and preparation [57]. The activity of sGC may also be altered by redox state, the oxidized form being unresponsive to NO [25]. Thus, the intensity of cyclic nucleotide signalling in pathological tissue remains unclear. While cGMP levels likely result from various states of the synthesis machinery, PDE activity is another crucial determinant of the cyclic nucleotide responses (see Section 3).

#### 2.3.2. Therapies to Stimulate the cGMP Pathway in PAH

In addition, alterations in pathways upstream of these systems are also important features in PAH. Several strategies have been employed to stimulate these pathways in an attempt to boost cyclic cAMP or cGMP in pulmonary circulation, yielding important progress in the management of PAH [9].

##### Promoting the NO–sGC Axis

The release of NO by the pulmonary endothelium is an important stimulator of cGMP synthesis in the PASMCs. Studies in mice demonstrated that the enzyme endothelial nitric oxide synthase (eNOS) is crucial in attenuating PA pressure and resistance [58]. The perturbation of NO bioavailability is a hallmark of PAH, but it is also present in other cardiovascular disorders as part of the endothelial dysfunction affecting various blood vessels [25,59]. Decreased eNOS expression [60] and low NO levels [61,62] have been observed in patients with PAH. Inhalation of NO at low doses has been approved to dilate PA and lower pulmonary vascular resistance in PAH patients [63]. However, long-term NO therapy is limited by the short half-life (15–30 s) of NO, which requires continuous inhalation and special delivery devices [64]. Currently, inhaled NO remains a standard method in evaluating acute vasoactive response during right heart catheterization testing as part of the diagnosis of PAH patients. NO is also used in pediatric medicine to treat PH associated with congenital heart disease [65,66] or in persistent PH of the newborn. More recently, the sGC stimulator, riociguat (Bay 63-2521), was developed as an alternative to stimulate cGMP [57,67]. This pharmacological class not only directly stimulates sGC but also sensitizes it to NO [68]. Riociguat showed curative effects in PAH animal models by partially reversing right heart hypertrophy and structural remodelling of the lung vasculature in PH models induced by CHx in mice or by monocrotaline (MCT) in rats [57]. Riociguat demonstrated efficacy in PAH patients and was approved to treat PAH and chronic thromboembolic pulmonary hypertension [9,25,67].

##### Promoting the NP System

Unlike NO, circulating BNP is elevated in PAH patients due to RV overload and dysfunction [69]. Genetic disruption of ANP or NPR-A receptor in mice demonstrated increased susceptibility to PH [70,71]. Infusion of BNP was well tolerated in PAH patients, but it did not significantly improve pulmonary hemodynamics. However, BNP potentiated the acute pulmonary vasodilator effect of the PDE5i sildenafil [72]. Because the circulatory half-lives of ANP (2–4 min) and BNP (~20 min) are short, both agents require continuous delivery and are thus difficult to use clinically. Intravenous administration of nesiritide, a recombinant BNP analogue, improved hemodynamic status in patients with post-capillary PH, but not with pre-capillary PH [73]. Drugs that include inhibitors of the breakdown of NPs by neutral endopeptidase neprilysin, such as ecadotril or sacubitril, have yielded promising results in animal models when combined with another pharmacological intervention, such as PDE5i or angiotensin receptor antagonist [74,75]. Such combinations would need further translational investigations. Another neprilysin inhibitor, racecadotril, was tested in patients with PAH and exhibited favourable short-term hemodynamic and tolerability [76]. These results warrant further investigation into the use of neprilysin inhibitors, possibly in combination with other therapies, including cGMP-elevating agents or PDE5i.

#### 2.3.3. Therapies to Stimulate the cAMP Pathway in PAH

The impairment of cAMP generation is consistent with the depression of the PGI_2_ pathway. PGI_2_ is a paracrine mediator produced in the endothelium by the PGI_2_ synthase, following processing of arachidonic acid by cyclooxygenases, COX. PGI_2_ and thromboxane A_2_ (TxA_2_), another arachidonic derivative, are important regulators of vascular tone in PA, although they have opposite effects [77]. PGI_2_ can bind to the prostaglandin I, IP receptor, which is coupled to G(α)s and to the cAMP pathway. The associated signalling in smooth muscle cells evokes vasodilation, while it inhibits PASMCs proliferation and platelet aggregation. TxA_2_ evokes vasoconstriction and platelet aggregation. Increased pulmonary vascular resistance is thought to result from an imbalance in biosynthesis of PGI_2_ and TxA_2_ in the pulmonary vascular bed, where the influence of TxA_2_ is stronger [78,79]. In fact, the expression of PGI_2_ synthase is reduced in small and medium-sized PAs in PAH patients [80]. Additionally, the response of AC to PGI_2_ is reduced in hypoxia-induced rat PA [81]. Alternatively, enhanced cAMP hydrolysis by PDEs, as detailed in the following sections, may also be an underlying cause of the deteriorated response to PGI_2_ in pathological pulmonary vasculature. Supplementation of PGI_2_ has been a strategy to improve the deteriorated IP receptor signalling in PAH patients. Infusion of epoprostenol, a synthetic PGI_2_–sodium salt, was the first therapy specifically approved for PAH. It significantly improves hemodynamics and long-term survival [82,83,84]. Continuous administration of PGI_2_ or epoprostenol is challenging due to its short half-life (2–3 min). Several PGI_2_ analogues (called “prostanoids”) have been developed with extended half-life and increased bioavailability: treprostinil, beraprost, and iloprost [9,85]. However, the constraints caused by their mode of administration (i.v. or s.c. infusion) and adverse effects complicate their use in routine. Side effects include pain at the site of injection and typical prostanoid-related symptoms such as headache, flushing, diarrhea, and jaw pain. As an alternative, selexipag was developed as an orally available, selective IP receptor agonist [9,86]. The active metabolite ACT-333679 (previously known as MRE-269) shows further selectivity for the IP receptor [86]. Selexipag was approved to treat PAH based on the beneficial hemodynamic effects [87] and the significant reduction in morbidity/mortality events in PAH [88].

## 3. Exploring PDE Families in Pulmonary Arteries and Relevance in PAH

### 3.1. Overview of PDE Families

Myriads of PDE isoforms are expressed in PA tissue (Table 2). In general, the total cGMP-PDE activity is much higher than the cAMP-PDE activity [42,52,89]. In PASMCs, PDE1, PDE3, and PDE4 are major enzymes that hydrolyze cAMP. In control patients, cAMP-PDEs activity ranked as follows: PDE4 > PDE3 > PDE1 > PDE2 [52]. As to cGMP-hydrolysis in PASMCs, most activity is ascribed to PDE5, and, to a lesser extent, to PDE1 [40,42,89]. More recently discovered PDEs, such as PDE7, PDE8, PDE9, PDE10, have been less characterized due to the lack of pharmacological or biochemical tools.

PDE activities for cAMP were found to increase in PASMCs from patients with PAH [52]. In PASMCs from PAH patients, PDE1 and PDE3 activities were higher, ranking as follows: PDE3 > PDE1 ≥ PDE4 > PDE2 [52,89,90]. In particular, the protein amounts of PDE1A, PDE1C, and PDE3B exhibited the most striking changes (Table 4) [39,52]. In animal models, cAMP-PDE activity measured in isolated PA was also higher in PH compared to control groups [39]. Likewise, cGMP-PDE activity appeared higher in PASMCs from PAH donors [52]. Nevertheless, tissues from rats exposed to CHx yielded inconsistent results: activity was higher in isolated PA, whereas it remained unchanged when measuring in the whole lung [39,56]. However, individual changes in specific cGMP-PDEs, PDE5 in particular, were reported (Table 4).

Pharmacological inhibition of PDEs is an efficient way to rescue cAMP responses, or to enhance cGMP levels [52,89,97] (Figure 2).

The following sections include a review of the basic research and clinical data about the respective contributions of PDE families in PAH. A summary of available data is provided in Table 4.

### 3.2. Phosphodiesterase 1 (PDE1)

#### 3.2.1. Enzymatic Properties

PDE1 is the unique PDE family that is activated by the Ca^2+^-calmodulin (CaM, a 16 kDa Ca^2+^-binding protein) complex, and is therefore often referred to as the “Ca^2+^/CaM-stimulated PDE” [102,103] (Table 2). Thus, this family potentially represents a unique regulatory link between cyclic nucleotides and intracellular Ca^2+^ signalling/handling, especially with respect to vascular tone. The PDE1 family is encoded by 3 genes: *PDE1A*, *PDE1B,* and *PDE1C*. The resulting isoforms, PDE1A and PDE1B, exhibit high affinity for cGMP, the latter being more selective for cGMP than the former. High affinity for both cAMP and cGMP is observed with PDE1C [102,103].

#### 3.2.2. Expression Pattern

Data from human and animal PA tissue and PASMCs show expression of PDE1A and PDE1C, while PDE1B is less abundant [41-43,51,52]. PDE1-related cAMP hydrolysis can be detected in PA in various species and is particularly prominent in human proliferative PASMCs [39,42,52]. This is consistent with the notion that PDE1C is highly expressed when proliferating smooth muscle cells have switched to a synthetic phenotype, as opposed to the contractile phenotype [103]. Nevertheless, only slight Ca^2+^-CaM-stimulated, cGMP- and cAMP hydrolyzing activities were detected in cytosolic and particulate fractions of human PA [40] and were ascribed to PDE1. Contribution of PDE1 to cGMP-hydrolysis in PA is limited compared to that of PDE5 in the absence of Ca^2+^-CaM [40,104].

In a CHx-induced rat model, it was observed that PDE1 contributed to increased cGMP-PDE and cAMP-PDE activities in the main PA [39]. PDE1A and PDE1C expressions were higher in PASMCs from idiopathic PAH (iPAH) patients compared with healthy donors, both at the mRNA and protein levels [43,52]. In addition, the immunolabelling signal for PDE1C was particularly prominent in PA from iPAH patients. In animal models, however, upregulation of PDE1 isoforms in tissue was less striking, with only the expression of *PDE1A* mRNA in PASMCs being significantly increased [43]. Interestingly, another study showed that in a cold-induced PAH model in rats, protein expression of PDE1C in PA was increased, whereas PDE1A expression was unaffected [41]. Although the quantification of protein expression in the lung vasculature was not reported, this may indicate that the regulation of PDE1 isoforms may vary among species and models. On the whole, these data suggest that PDE1 activity increases in pathological PA; however, the identities of underlying isoforms may vary between species and models, with PDE1C as likely the most relevant candidate in human PAH.

#### 3.2.3. Functional Role and Therapeutic Potential

It was reported that treatment with PDE1C-targeted siRNA both enhanced cAMP accumulation and inhibited cellular proliferation to a greater extent in PASMC from PAH patients than in controls [52]. This is consistent with the role played by PDE1C in promoting vascular smooth muscle cell proliferation in other diseases, where vessel wall remodelling was involved [5,103,105,106]. Accordingly, PDE1 inhibitors (PI79, vinpocetine) slowed proliferation in human and rat PASMCs [43,51]. PDE1 was shown to promote proliferation and inhibit apoptosis through various mechanisms (reviewed in [5]). In addition, PDE1 inhibitors have been reported to exert a vasorelaxant effect on mouse PA, rat lung, and lamb PA [43,51,107].

Preclinical therapeutic assessment of another PDE1 inhibitor (8-methoxymethyl 3-isobutyl-1-methylxanthine, 8MM-IBMX) was performed in three animal models of PAH, namely MCT-treated rats, cold-induced PAH rats, and CHx mice [41,43]. Pharmacological treatments were administered for 1 to 2 weeks once the disease was established, following a curative-like strategy. PDE1 inhibitor resulted in reduced PAP, reversed remodelling of the lung vasculature, and a reduction in RV hypertrophy. In addition, 8MM-IBMX was reported to suppress macrophage infiltration and superoxide production [41]. Taken together, these findings indicate that upregulation of PDE1, especially PDE1C in humans, plays a role in the structural remodelling process underlying PAH and, thus, offers a novel therapeutic target.

#### 3.2.4. Perspectives and Limitations

Although vinpocetine [108] and 8-MM-IBMX [106,109] are commonly used PDE1 inhibitors, the selectivity of these compounds for PDE1 is moderate. For example, vinpocetine also inhibits several sodium channels [110] and inhibits NF-κB-dependent inflammation via an IκB-dependent but PDE-independent mechanism [111] and is more selective for PDE1A and PDE1B in comparison with PDE1C [112]. 8-MM-IBMX has increased selectivity for PDE1 over other PDEs by 30–50 fold [109]. Therefore, pharmacological effects should be interpreted cautiously. However, some promising PDE1 selective inhibitors have emerged. For example, Lu AF41228 and Lu AF58027 can induce vasodilation of the mesenteric artery and lower blood pressure [113]. Furthermore, dioclein and ITI-214 can enhance cardiac function acutely (reviewed in [5]). As previously discussed, because expression patterns in animal models may not accurately mimic those in patients, the translational value of in vivo studies remains uncertain. Data from human tissue models or clinical studies are clearly needed to ascertain the therapeutic potential of recent PDE1 inhibitors in PAH.

To date, no clinical trial has been conducted to test the efficacy of PDE1 inhibitors in PAH [114]. ITI-214 was recently entered into clinical trials (phase 1/2, NCT03387215) for the treatment of heart failure, providing an interesting opportunity to evaluate the safety and efficacy of PDE1 inhibitors in cardiovascular disorders [115]. Considering that PDE1A and C are expressed in the RV [116] and that their expression rises in the failing ventricle [5], PDE1 inhibition may promote cardiac effects in the hypertrophied RV of PAH patients. Whether these cardiac and vascular effects would translate into beneficial (improved RV function) or deleterious (arrhythmias) outcomes remains uncertain and needs to be investigated.

### 3.3. Phosphodiesterase 2 (PDE2)

#### 3.3.1. Enzymatic Properties

PDE2 hydrolyzes both cGMP and cAMP with similar maximal velocity rates and relatively high Km values [117,118]. PDE2 contains two homologous GAF domains tandemly arranged in its regulatory regions. The GAF-A domain of PDE2 is involved in dimerization, while the GAF-B domain binds cGMP. The latter triggers conformational changes, increasing cAMP hydrolysis up to 30-fold [117,118,119]. Thus, PDE2 is unique in that it is the only cGMP-activated, cAMP-hydrolyzing PDE. PDE2A GAF-B is moderately selective with around 20-fold preference for cGMP (K_D_ < 10 nM) over cAMP [119,120]. PDE2 isozymes are encoded by a single gene, *PDE2A,* containing three known variations (*PDE2A1*, *PDE2A2*, *PDE2A3*). The resulting isoforms differ from one another in their N-terminal domain, which likely influences the subcellular localization of the proteins. Specifically, PDE2A1 is more soluble while PDE2A2/A3 are preferentially membrane-associated [102].

#### 3.3.2. Expression Pattern

PDE2 mRNA and activity can be detected in PA from various species, but they exhibit lower levels compared to PDE3, PDE4, and PDE5 [39,42,47,52]. A clear cGMP-stimulated cAMP hydrolyzing activity was detected in rat PA [46]. This result did not differ from that obtained in “denuded” PA (with the endothelial layer disrupted), indicating the expression of PDE2 in the media layer. Overall, substantial data support the existence of PDE2 protein and its activity in PASMCs. By contrast, PDE2 expression in the media of other vascular beds remains elusive (further discussed in a study by Wang et al. [121]).

Studies on the expression of *PDE2A* in PASMCs from PAH patients yielded variable results, reporting either up-regulation [52] or down-regulation [47] compared to control samples. PDE2-related activity was, however, not significantly altered [52].

In addition, PDE2 expression in ECs, including in PA, was also documented [48,122].

#### 3.3.3. Functional Role and Therapeutic Potential

Pharmacological inhibitors of PDE2, such as erythro-9-(2-hydroxyl-3-nonyl) adenine (ENHA, IC50 = 0.8 μM) and Bay 60-7550 (IC_50_ = 4.7 nM), have been used to explore the contribution of the enzyme to physiological and pathological processes [123,124,125]. EHNA can increase forskolin-induced cAMP accumulation in PASMCs [52], demonstrating PDE2 activity. In the study by Bubb et al. [47], the effect of Bay 60-7550 was investigated alone or in combination with cyclic nucleotide stimuli in various experimental systems. As summarized earlier in this review (Section 2.2), responses to ANP and NO are typically ascribed to cGMP, while treprostinil is mediated by the cAMP pathway. The proliferation of PASMCs from PAH patients was reduced by Bay 60-7550, an effect potentiated in the presence of ANP, NO, or treprostinil. As to vascular reactivity, EHNA can reverse hypoxic vasoconstriction ex vivo in perfused rat lungs [46] and exhibits moderate vasorelaxant effects [42]. In PA isolated from rats exposed to CHx, Bay 60-7550 produced a dose-dependent relaxation at the micromolar range [47]. Furthermore, Bay 60-7550 (0.1 µM) potentiated vasorelaxant responses to ANP or treprostinil. In vessels from normoxic rats, responses to a NO donor were also potentiated by Bay 60-7550. However, responses to treprostinil remained unchanged. Paradoxically, the authors found that PDE2A expression was down-regulated in isolated PA tissue from CHx-induced rat models [47]. Overall, these data show that PDE2 is likely involved in PA contractile reactivity and suggest that PDE2 differentially mitigate various cyclic nucleotide pathways in health or disease. Nevertheless, the specific contributions of either endothelial or smooth muscle PDE2 remain unclear and need to be further defined. Genetic ablation of *PDE2A*, restricted in a cell-type-specific manner, will possibly address this question. PDE2 plays a role in endothelial cells (ECs) as a regulator of barrier integrity and angiogenesis [5,48]. Inhibition of PDE2 is beneficial in reducing endothelial permeability triggered by tumour necrosis factor-α or H_2_O_2_ [48,126]. These findings are consistent with the general roles of PDE2 demonstrated in ECs, hindering endothelial barrier function (reviewed in [118]).

Most importantly, Bay 60-7550 prevented the development of both hypoxia- and bleomycin-induced PH in mice without influencing the systemic arterial pressure [47]. Interestingly, this protective effect on RV systolic pressure and on RV hypertrophy was decreased in NPR-A (GC-A) knockout animals, but not in mice receiving the inhibitor of nitric oxide synthases L-NAME. These results suggest that PDE2 inhibition exerts a salutary effect in PH by preferentially promoting cGMP derived from the NP pathway, rather than the NO pathway, a mechanism similar to that reported in cardiac myocytes [127]. Nevertheless, the PDE2 inhibitor alone, administered from day 14 once PH was established, was not effective in curing PH. When administered in combination with the neutral endopeptidase inhibitor ecadotril (to enhance endogenous NPs levels), treprostinil, inorganic nitrate (NO donor), or the PDE5i sildenafil, Bay 60-7550 produced a significantly greater reduction in the salient features of the model compared with monotherapy with each compound alone. Therefore, one may hypothesize that PDE2 inhibition could serve as an effective adjuvant therapy to bolster both cGMP and cAMP signalling in pulmonary circulation, synergizing with pharmaceuticals such as NPs, PGI_2_ analogues, or even inorganic nitrates.

#### 3.3.4. Perspectives and Limitations

The unique molecular properties of PDE2 enable crosstalk between both cAMP and cGMP pathways, with potential pathophysiological and therapeutic consequences [118]. Interestingly, an additive effect of PDE2 inhibition on top of PDE5i was observed against PASMCs proliferation or in preventing in vivo development of PAH [47]. The reason for this may be that the inhibition of PDE5 exacerbates PDE2 activity by promoting a rise in cGMP and binding to the GAF-B domain. This may potentially explain how the inhibition of both enzymes may bolster cyclic nucleotide pathways. The safety of such a combination would need to be carefully monitored, as it may increase the risk of hypotension; however, no change in mean arterial pressure was observed in the mouse CHx model used in the study by Bubb et al. [47]. Considering the various roles that PDE2 is thought to play in ECs (promoting angiogenesis and barrier leakage) and in the hypertrophied cardiomyocyte (reviewed in [118]), the efficacy and safety of PDE2 inhibition in pulmonary circulation require further exploration using different models. Because global genetic knockout of *PDE2A* is lethal at the embryonic stage, cell-type-specific models would be valuable tools for further investigations. In conclusion, PDE2 may be an interesting target to consider in PAH, although mechanisms impacted by enzyme inhibition are still uncertain and may impact various functions. Furthermore, a combination with other PDE inhibitors or therapies that enhance cyclic nucleotides would be more likely to reach efficacy endpoints.

Several PDE2 or mixed PDE2/5 inhibitors developed to treat migraine and cancer have failed to reach a significant stage in clinical trials, primarily due to safety concerns (reviewed in Baillie et al. 2019) [35]. Recent small molecules have been investigated: PF-05180999 (Pfizer) was terminated in a Phase I clinical trial due to safety concerns in patients with migraines [35]. TAK-915 (NCT02461160), which recently completed a Phase I trial.

### 3.4. Phosphodiesterase 3 (PDE3)

#### 3.4.1. Enzymatic Properties

The PDE3 family is coded by two genes, namely *PDE3A* and *PDE3B*, which show high affinity for both cAMP and cGMP [117] (Table 2). A lower V_max_ for cGMP compared to cAMP (approximately 4 to 10-fold) makes cGMP a competitive inhibitor for cAMP hydrolysis [128]. Therefore, PDE3 is often referred to as the “cGMP-inhibited, cAMP-hydrolyzing PDE”.

A variety of compounds demonstrate PDE3 inhibiting properties [129], and some of these were used at selective concentrations to assess PDE3 activity in PA tissue and cells (e.g., motapizone, milrinone, cilostamide, cilostazol, etc.) [35,39,40,42,117]. PDE3 activity contributes to the largest portion of cAMP hydrolysis in PA and PASMCs, either in control samples (30–60%) or in patients as well as PH models (around 40–50%) [39,40,42,52]. PDE3 also contributes to cGMP hydrolysis [40].

#### 3.4.2. Expression Pattern

Data obtained in PASMCs from human patients showed that both PDE3A and PDE3B are expressed in human PASMCs. Both mRNA levels are upregulated in idiopathic PAH patients, but only PDE3B protein expression is increased [52].

PDE3 activity increases in PASMCs from iPAH patients and in large, intralobar PA from rats exposed to CHx, which can partly explain the total increase in cAMP-related PDE activity in these samples and dovetails well with the changes in PDE3 expression [39,49,52]. PDE3 activity is also prominent in PAECs, where PDE3 inhibition with motapizone can protect against the H_2_O_2_-induced increase in endothelial permeability [48].

#### 3.4.3. Functional Role and Therapeutic Potential

The vasorelaxant effects of PDE3 inhibitors have been extensively explored in the pulmonary vasculature from various species, including humans. PDE3 inhibitors effectively relax ex vivo precontracted PA [40,42,130] and potentiate cAMP-mediated vasorelaxant responses [131]. For example, milrinone rescued the cAMP responses in the CHx context, where cAMP signalling was otherwise blunted [131]. This result is consistent with increased PDE3 expression limiting the cAMP activity in pathological PA. Similar effects were reported as to the capacity to drop pulmonary vascular resistance in perfused lungs or in vivo [132,133,134]. These properties potentiate or synergize with responses to approved PAH therapies, such as PDE5i or inhaled PGI_2_ [132]. PDE3 inhibition also enhances cAMP levels and produces an anti-proliferative effect in PASMCs, particularly in cells from patients with iPAH [52]. Taken together, data show that PDE3 is a master repressor of cAMP levels in PA under both physiological and pathological conditions. In vivo, PDE3 inhibitors (cilostazol, cilostamide) attenuate RV systolic pressure and RV hypertrophy in the MCT-induced PH model, but not in CHx-induced models [51,91,92], unless combined with another cAMP-elevating agent [51]. Likewise, nebulization of a dual PDE3-PDE4 inhibitor improved the MCT-induced pulmonary vascular damage and RV hypertrophy [135].

#### 3.4.4. Perspectives and Limitations

Consistent with ubiquitous PDE3 expression through smooth muscle cells [27], the vasodilatory effects of PDE3 inhibitors, or other mixed and non-selective inhibitors, extend to systemic vasculature and promote hypotension. This is a limitation to straightforward translation of these compounds for use in clinics [133,136].

In addition, PDE3 inhibitors such as milrinone, like other inotropic agents, are only used for the treatment of patients with acute heart failure, while chronic use is limited due to safety concerns [137]. Likewise, milrinone produces inotropic effects on the RV in infants with persistent PH of the newborn, where it may be useful as an adjuvant therapy in addition to NO [138,139]. Nevertheless, PDE3 inhibition in the RV may potentially turn arrhythmogenic, as suggested by non-clinical data obtained in a porcine model [140]. Such systemic effect of PDE inhibition may be circumvented by titrating the lowest effective dose according to the pulmonary versus systemic selectivity and by using the respiratory route of administration of nebulized inhibitors, as investigated by Schermuly et al. [141].

In summary, PDE3 inhibitors administered alone showed modest advantages in experimental PAH treatment and present potential safety issues due to cardiac and systemic pharmacodynamics. To date, there has been no clinical trial testing PDE3 inhibitors for PH in chronic settings. In practice, such pharmacological approaches are currently relevant in an acute setting to address PH and RV dysfunction during cardiac surgery [142] and pediatric intensive care units, such as persistent PH of the newborn [138,139,142].

### 3.5. Phosphodiesterase 4 (PDE4)

#### 3.5.1. Enzymatic Properties

PDE4 is well characterized by its substrate selectivity for cAMP and its sensitivity to inhibitors such as rolipram or roflumilast [5,35]. PDE4 isozymes comprise four subfamilies (4A, 4B, 4C, and 4D), each encoded by a distinct gene, resulting in more than 25 PDE4 isoforms in humans [143]. Based on the presence of upstream conserved regions domains 1 and 2 (UCR1 and UCR2), PDE4 isoforms can be divided into four categories: long isoforms harbouring both UCR1 and 2, short isoforms having only UCR2, super-short isoforms with a truncated UCR2, and dead-short isoforms that lack both UCR domains and have a truncated catalytic domain [143]. Various features of the protein, including UCR modules, define many post-translational properties of the enzyme, such as regulation by phosphorylation, dimerization of long forms, and anchoring (or targeting) to subcellular structures (reviewed in [5,143]).

#### 3.5.2. Expression Pattern

PDE4 activity can be revealed by using various selective inhibitors such as rolipram (IC_50_: 0.05–2 μM), Ro 20-1724 (IC_50_: 1–2 μM) [144], or roflumilast (IC_50_: 0.4 nM). PDE4 is generally well detected in PA tissue, but at variable proportions. PDE4 contributes less than half of PDE3 cAMP-hydrolyzing activity [40] in humans but is almost as important as PDE3 in bovine PA [42]. Although cAMP-PDE activity is globally increased in PASMCs from PAH patients, the PDE4 activity does not contribute to this, because PDE4 activity decreases in PAH patients (30% of total) compared to controls (around 45%); meanwhile, there was an increase in activities ascribed to PDE1 and PDE3 [52]. Additionally, PDE4 is clearly a PDE responsible for cAMP hydrolysis in the endothelium, as demonstrated in ECs from porcine PA by using rolipram [48]. A recent study showed that EC was one major cell type expressing *PDE4B* in the lung. This isoform was the only one whose expression was significantly upregulated in several models of experimental PH and in PH patients [93].

#### 3.5.3. Functional Role and Therapeutic Potential

In human PA, PDE4 inhibitors alone exert only poor vasorelaxant effects unless PDE3 is simultaneously inhibited [40]. This suggests that the overlapping activities of both enzymes are responsible for regulating vascular tone. By contrast, rolipram potently relaxes rat PA [42]. The discrepancy between these results may originate from the species used or the agonist used to obtain precontraction of the arterial rings (PGF2α vs. phenylephrine). High concentration of rolipram can relax the rat PA contracted by acute hypoxia in the perfused lung [51]. PDE4 inhibitors also exhibit anti-proliferative effects in rat and human PASMCs [44,51,52]. These effects, however, were not increased in PASMCs from PAH patients compared with controls [52]. Taken together, the data show that although PDE4 is a substantial contributor to cAMP hydrolysis in PASMCs, limiting vasorelaxation and promoting cell proliferation, it is not significantly altered in PAH.

On the contrary, PDE4 located in PAECs may be a more likely contributor to PA tissue remodelling. PDE4 inhibition by rolipram was shown to limit endothelial permeability induced by H_2_O_2_ [48], synergizing with PGE1. More specifically, Xing et al. provided data showing that PDE4B participates in experimental PH (see below). This process involves attenuation of the PKA-CREB-BMPR2 axis as a possible mechanism [93].

In vivo studies conducted to test the efficacy of PDE4 inhibition in PH models yielded mixed results. In the rat CHx model, rolipram attenuated neither mPAP, RV hypertrophy, nor the distal vessel muscularization, except when combined with iloprost or a PDE3 inhibitor [51]. Another PDE4 inhibitor, roflumilast, used at high dose (1.5 mg/kg), prevented both MCT- and CHx-induced PH phenotype in rats [94]. Moreover, roflumilast partly reversed the established PH phenotype induced by MCT (curative protocol) [94]. A lower dose (0.5 mg/kg) was only effective in preventing MCT rather than CHx-induced PH phenotype [94].

More recently, curative effects of roflumilast at high dose (2 mg/kg/day) were confirmed in the SuHx model in mice [93]. The same group demonstrated that genetic invalidation of *PDE4B* protected mice against salient alterations evoked by the SuHx model. Interestingly, the salutary effects of roflumilast were not observed in *PDE4B^−/−^* mice, confirming that the pharmacological action of roflumilast could be ascribed to selective inhibition of PDE4B isoform. Likewise, another PDE4B-selective inhibitor (BI 1015550, 3.24 mg/kg/day, per os) improved the SuHx-induced rat PH phenotype [93].

Because endothelial *PDE4B* was one major PDE isoform that was upregulated in experimental PH, Xing et al. investigated mice with *PDE4B* deletion restricted to ECs, using expression of Cre recombinase under the control of the *Tek* gene promoter. It was found that *PDE4B^EC^^−/−^* mice showed less severe PH phenotype in SuHx models, compared to *PDE4B^EC+/+^* “floxed” mice. In this context, the authors examined the EndoMT process, which is characterized by a decrease in endothelial markers and a concomitant increase in mesenchymal markers such as the α-smooth muscle actin gene (*Acta2*) expression. While EndoMT increased in the SuHx model, it was limited by the *PDE4B* ablation. Taken together, while PDE4 is quantitatively an important cAMP-hydrolyzing PDE in pulmonary circulation, PDE4B in ECs may be a potential target for future therapeutic development.

#### 3.5.4. Perspectives and Limitations

These advances need to be translated into human cells and tissues, while the PDE4B inhibitor BI 1015550 has so far only been tested in patients with idiopathic fibrosis [145,146]. Supported by the rationale that activity spectra of PDE4 and PDE3 overlap, combined inhibition of PDE3 and PDE4 provided efficacy when administered with iloprost in the rat MCT and CHx models [51,147]. Data obtained from PH patients show that acute tolafentrine (a dual PDE3/PDE4 inhibitor) could serve as a useful adjuvant to iloprost [148]. Nevertheless, further investigations are necessary to explore the long-term potential of this therapy. One may speculate that similar limitations may apply to PDE4 inhibitors as to PDE3 inhibitors, although the level of expression of PDE4 is much lower than PDE3 in human cardiac tissue, thereby limiting the potential inotropic and proarrhythmic effects of a possible PDE4 inhibition-based therapy [24].

Conversely, the well-characterized anti-inflammatory action of PDE4 inhibitors may complement therapies acting more specifically on tissue remodelling [5]. More specifically, PDE4 is expressed in immune cells and promotes cell activity by lowering cAMP levels. In addition, chronic inflammation is an accepted component of various forms of PH (review in [149,150]), although its causal role in the pathogenesis remains unclear. Inflammation primarily manifests as the infiltration of immune cells into vascular lesions and of increased levels of circulating pro-inflammatory mediators. While PDE4 inhibition limited the increase in some cytokines in MCT-induced PH [94], data in other relevant PH models are needed.

The clinical development of PDE4 inhibition has been hindered by adverse side effects, mainly in the gastrointestinal area [5]. Off-targeting of PDE4D located in the area postrema may be responsible for emetic symptoms [5]. Targeting PDE4B more selectively than PDE4D may increase the tolerability of PDE4 inhibitors therapies by preserving the efficacy of inhibition of the most relevant PDE isoform. The use of isoform-selective inhibitors, such as the anti-fibrotic nerandomilast (BI 101555), may offer therapeutic insight through future dedicated clinical investigation.

### 3.6. Phosphodiesterase 5 (PDE5)

#### 3.6.1. Enzymatic Properties

PDE5 is a cGMP-specific PDE, and is coded only by one gene, *PDE5A*, that can yield three variants (PDE5A1, PDE5A2, and PDE5A3). Structurally, the PDE5 N-terminal region contains two allosteric cGMP-binding domains (GAF-A and GAF-B) and one phosphorylation site (at Ser92 in bovine and Ser102 in human) [151,152]. High-affinity (Kd < 40 nM) cGMP binding occurs only to the GAF-A domain, which stimulates enzyme activity 9 to 11-fold [153]. The role of GAF-B is currently unclear. Phosphorylation at Ser92 by PKG or PKA not only activates the catalytic function but also further increases the allosteric cGMP-binding affinity, thus working as positive feedback to mitigate cGMP levels [102,151].

#### 3.6.2. Expression Pattern

Compared with other PDEs, PDE5 has the highest expression and contribution in GMP hydrolysis in the lungs, pulmonary arterial vessels, and PASMCs, both from animals [42,104,154] and humans [40,52]. PDE5 expression and activity are strongly increased in the lungs and/or PA from animal models or PAH patients [49,50,89]. Expression markedly localizes in the thickened media layer [89]. This suggests that PDE5 represents a marker of vascular remodelling in PAH development. Furthermore, the expression of PDE5 was reported to increase in the hypertrophied RV compared with the normal RV both in PAH patients and MCT-induced PAH rat models [155], indicating that cardiac pharmacodynamics may also account for the therapeutic action of PDE5i.

#### 3.6.3. Functional Role and Therapeutic Potential

Several studies addressed the role of PDE5 in PAH, providing a compelling basis for the treatment of PH [9,35]. Ex vivo studies have consistently shown that PDE5i relaxes PA in both humans [40,156] and animal models of PAH [56,74,104]. Furthermore, the inhibition of PDE5 increases cGMP levels in a concentration-dependent manner, producing anti-proliferative and pro-apoptotic effects in PASMCs from patients with PAH. These effects can be augmented synergistically or additively by NO donors or NP analogues [89,156]. Moreover, in vivo studies show that PDE5i not only can prevent the development of various PAH models [74,157,158,159], but also partially reverse the established remodelling of the PA wall [50,158]. This translates into a decrease in mean PAP or RV systolic pressure, a reduction in RV hypertrophy, and mitigation of PA remodelling or of distal vascular muscularization, without affecting systemic blood pressure. Regarding the cardiac effects, acute inhibition of PDE5 in isolated hypertrophied RV exerts positive inotropism, which is mediated by increased cAMP levels, resulting from the inhibition of PDE3 activity by increasing cGMP [155].

The beneficial effects of PDE5 inhibition obtained in preclinical models warranted the assessment of three compounds in clinical trials: sildenafil, tadalafil, and vardenafil. These inhibitors have demonstrated efficacy in improving clinical endpoints such as hemodynamics, six-minute walk distance, and quality of life in PAH patients [160,161,162,163]. In the long term, PDE5i improve the time-to-clinical worsening and mortality [164].

#### 3.6.4. Perspectives and Limitations

PDE5i are currently a cornerstone of the management of PAH patients (see Section 1.3 above), as a part of first-line therapy, regardless of risk status [4,6]. PDE5i are generally well-tolerated, with common systemic side effects ascribed to PDE inhibition in various tissues. These effects include headaches, flushing, vision problems, gastrointestinal discomfort, and muscle and joint pain [165,166].

Mechanistically, the efficacy of PDE5i relies on the status of cGMP synthesis by either the NO or NPs pathway [25,97,167]. In smooth muscle cells, PDE5 is distributed to specific subcellular targets and participates in the compartmentalization of both branches of the cGMP signalling. The range of cellular mechanisms targeted by PDE5i encompasses the targets of PKG [17,19], including Ca^2+^ handling and signalling mechanisms, ion channels, and the RhoA pathway [168,169]. As previously mentioned, the inhibition of PDE5 may lead to the inhibition of PDE3 or the activation of PDE2, thereby establishing a complex cross-talk between cAMP and cGMP pathways. Nevertheless, an accurate description of cellular compartments involving PDE5 and other PDEs is still lacking and needs to be addressed by further research. In particular, the preferential contribution of PDE5 to either membrane- or cytosolic-mediated cGMP pools is uncertain in PA. The combination of PDE5i with riociguat promotes vasodilation, suggesting that PDE5i potentiates cGC activity [170] and that PDE5 connects with the NO–sGC–cGMP axis. Moreover, PDE5i synergize with NP-enhancing drugs, indicating that PDE5 also mitigates the NP receptor-generated cGMP pool [74]. Thus, PDE5 is likely to be involved in the control of global cGMP in the smooth muscle cells.

The role of PDE5 as a master controller of cGMP limits the use of further combinational therapy with other cGMP enhancers in clinics. This would lead to excesses in pharmacodynamic responses. The association of a PDE5i with riociguat is therefore contraindicated due to the risk of hypotension and other side effects [171]. In addition, the selectivity of PDE5i is relative, and other PDE isoforms may be affected, depending on the reached plasma concentration [166]. Recently, novel, highly selective PDE5i have been reported, such as dihydroquinoline-2(1H)-ones [99] and DDCI [98,172]. These new compounds may offer improved selectivity compared to sildenafil or tadalafil, improving the tolerability of PDEi among patients.

### 3.7. Phosphodiesterase 9 (PDE9)

#### 3.7.1. Enzymatic Properties

PDE9 was discovered in the late 1990s and is coded by a single gene, *PDE9A* [173,174,175]. *PDE9A* was reported to encompass more than 21 splice variants, possibly displaying differential subcellular localizations and properties [176,177]. PDE9 is highly selective for cGMP hydrolysis with a Km of about 70–170 nM for cGMP and only 230 µM for cAMP [175]. Importantly, the affinity of PDE9 for cGMP is 40–170 times higher than two other cGMP-specific PDEs, PDE5 and PDE6, respectively. The maximum velocity of the catalytic site is similar to PDE5. All these properties make PDE9 a potential sensitive tuner of cellular cGMP signalling.

#### 3.7.2. Expression Pattern

The tissue distribution of PDE9 is ubiquitous, as PDE9 mRNA has been detected in many organs. The highest levels are reported in the brain, kidney, spleen, with relatively lower levels found in the lung [173,174,175], and the heart [174,175,178]. Studies accounting for the expression of PDE9 in the pulmonary vasculature are scarce. PDE9 was only mentioned to be expressed at the mRNA level in fetal rat and human PASMCs in a couple of reports [52,179]. Another report mentioned that PDE9 inhibition decreased leucocyte adhesion in arterioles [180]. Therefore, expression of PDE9 in ECs cannot be ruled out. Regarding PDE9 expression in the context of PH, a study reported that PDE9 mRNA was not altered in PASMCs from iPAH patients but was upregulated in Group III PH [52].

#### 3.7.3. Functional Role and Therapeutic Potential

To date, several selective PDE9 inhibitors have been characterized. Studies mainly focused on their ability to increase cGMP in the brain and rescue cognitive function [181,182,183,184,185,186] (reviewed in [5,35]). Importantly, PDE9 was also explored as a therapeutic target in experimental heart failure. PDE9 expression in the myocardium was reported to increase in subjects with dilated cardiomyopathy and heart failure, or in cardiomyocytes submitted to a pro-hypertrophic stimulus [178]. Several studies demonstrated that PDE9 global ablation or pharmacological inhibition (PF-04449613, PF-04447943, CRD-733) can improve cardiac hypertrophy, diastolic dysfunction, and coronary microvascular rarefaction in models of hypertrophy and myocardium dysfunction [178,185,187]. In addition to the direct effects on cardiac myocytes, PDE9 inhibition may also produce pharmacodynamic effects on the vasculature. Acute PDE9 inhibition with PF-047499982 dose-dependently diminished systemic blood pressure and peripheral vascular resistance in a heart failure model in sheep, indicating a vasodilatory action [188,189,190]. Moreover, the PDE9 inhibitor decreased pulmonary arterial resistance, although not to the same degree as sildenafil [190]. Therefore, PDE9 inhibition may induce the relaxation of PASMCs in vivo, although direct evidence is still lacking. Although the selectivity indexes of PF-04449613 and PF-047499982 towards PDE9 are at least 40 and 300 times over other PDEs, a non-selective effect on other PDEs cannot be ruled out in these in vivo studies [182,191].

Like other PDEs, the contribution of PDE9 to cGMP compartmentalization has been explored [167]. By following a pharmacological approach combined with cGMP measurements, by means of an intracellular FRET-based biosensor, data by Zhang et al. suggested that PDE9 selectively regulates NO-related cGMP levels in rat aorta smooth muscle cells [192]. This conclusion is at odds with other results obtained from cardiac myocytes, since Lee et al. demonstrated that PDE9 mitigates only cGMP pathways related to NP rather than the NO pathways [178]. In addition, in sheep with heart failure, PF-047499982 produced additional improvement in promoting the hemodynamic effects of NPs when combined with neprilysin inhibition [188].

Altogether, while the expression and role of PDE9 in vasculature remain unclear, PDE9 inhibition may produce vasodilatory responses that could be useful in fine-tuning cardiac preload and afterload and may also oppose the progression of PH. Nevertheless, one study reported that *PDE9A*-deficient mice were not protected from CHx-induced PH [101]. A recent study demonstrated that PDE9 expression increased in the lungs of aged mice and that treatment with PF-04447943 reversed the age-related increase in pulmonary vascular resistance. It also improved the vasodilatory response to sodium nitroprusside in isolated, perfused lung [187]. These data suggest that the physiological role of PDE9 may be more prominent in aged animals.

#### 3.7.4. Perspectives and Limitations

Several PDE9 inhibitors have been tested in humans, with early clinical trials, such as PF-04447943 (clinical trial Ib) [193] and BI 409306 [194]. Ongoing clinical trials are evaluating the therapeutic effects of CRD-740 (NCT05409183) and tovinontrine in heart failure (NCT06215586, NCT06215586). Despite possible promising hemodynamic properties, the usefulness of PDE9 inhibition in improving pulmonary vasculature or RV dysfunction remains unclear and warrants further non-clinical studies in order to support future clinical investigation in PAH patients.

### 3.8. Phosphodiesterase 10 (PDE10)

#### 3.8.1. Enzymatic Properties

PDE10 is coded by a single gene, *PDE10A*. PDE10 hydrolyzes both cAMP and cGMP, with a higher affinity for cAMP than for cGMP. V_max_, however, is 2–5-fold lower for cAMP compared to cGMP. Therefore, cAMP inhibits cGMP hydrolysis, thus making this enzyme a cAMP-inhibited cGMP PDE [38]. PDE10 harbours two GAF domains in the N-terminal region. In contrast to other PDEs, only cAMP can bind to the GAF domain, which activates PDE10 [195,196]. Papaverine has been used as a potent inhibitor of PDE10A, although it also inhibits PDE3 with a selectivity ratio < 10 [53,197].

#### 3.8.2. Expression Pattern

Expression of PDE10 was reported in human PASMCs, but no significant variation in transcript levels was reported in iPAH PASMCs [52]. Tian et al. explored PDE10 in the MCT-induced PH model in rats [53]. They found that PDE10 transcripts and protein levels were more abundant in PASMCs, PA, and lung isolated from rats with PH, compared to control rats. In particular, immunolabelling indicated a strong PDE10 signal in the remodelled media layer from iPAH donors and MCT-exposed rats [53].

By using 10 µM papaverine to reveal the activity of PDE10, surprisingly high levels were reported in PASMCs. Indeed, PDE10 contributed to 38% and 53% of cAMP hydrolysis in PASMCs from control and MCT rats, respectively. However, papaverine is probably not selective at this concentration, and activities of other PDEs, such as PDE3, may also be included in these measurements. Thus, PDE10 seems to be highly expressed in the remodelling PA media. However, further exploration of animal models and human tissues is needed to ascertain the contribution of PDE10 to cAMP activity.

#### 3.8.3. Functional Role and Therapeutic Potential

Papaverine was shown to increase cAMP levels and to activate downstream phosphorylation of the transcription factor CREB and exert an anti-proliferative effect on PASMCs isolated from MCT-exposed or control rats [53]. Papaverine and, importantly, other more recent synthetic PDE10 inhibitors were tested for the prevention of MCT-induced PAH [53,198,199]. Interestingly, these compounds decreased mPAP or RV systolic pressure, RV hypertrophy, PA thickness, and distal vascular muscularization [53,198,199]. The interpretation of the MCT model, however, is of limited translational value as it does not recapitulate all the severe features of PAH in humans [200]. Additional studies should be performed in other PH models to confirm the salutary effects of PDE10 inhibition.

In addition, PDE10 expression increased in mouse and human failing hearts, and PDE10A inhibition was found to improve cardiac remodelling and dysfunction [201]. It may be speculated that PDE10 inhibition would also protect against RV hypertrophy and fibrosis, although additional studies are required to verify this.

#### 3.8.4. Perspectives and Limitations

As discussed previously, the selectivity of papaverine for PDE10 is limited. Therefore, efforts using highly selective inhibitors of PDE10 may help demonstrate the translational potential in PH. Several PDE10 inhibitors have been developed in the central nervous system area, some being tested in schizophrenia, Huntington’s disease, and cancer [5,35,202]. Such growing diversity of selective PDE10 inhibitors may afford opportunities for potent pharmacological approaches to treat PAH. Nevertheless, because PDE10 is abundant in the brain, strategies that favour pulmonary and cardiac selectivity would be valuable to preserve the tolerability of future drug candidates.

### 3.9. Other PDEs

Other PDEs include PDE6, PDE7, PDE8, and PDE11, for which there is little information, if any, regarding their potential role in PA. The PDE6 family is restricted to the retina and works as a photoreceptor. Transcripts of PDE7A were reported to be more abundant in PASMCs from PAH patients [52] and rat models [53], whereas *PDE7B*, *PDE8A*, *PDE8B*, and *PDE11A* showed no significant changes in PASMCs from iPAH patients [52]. These PDEs are, for now, mostly studied in diseases of the central nervous system [5,35].

## 4. Future Directions and Limitations

### 4.1. Potential Systemic Adverse Effects of PDE Inhibition (e.g., Hypotension, Cardiac Effects)

There are several potential limitations in the use of PDE inhibitors in PH and PAH. These include redundancy of PDE activity, difficulty in selectively targeting discrete PDE isoforms in specific cell types, the lack of selectivity of pharmacological inhibitors, ubiquitous expression of some PDE isoforms, and the insufficient expression of others.

The localization of the PDEs expression across the vascular territories is an important determinant to consider. The success of PDE5i relies on their ability to selectively relax pulmonary circulation, while having little influence on the systemic blood pressure [56,96]. Cyclic-AMP PDEs, such as PDE3, may be more problematic as their inhibitors are potent vasodilators across the vascular system [5,38].

More generally, inaccurate inhibition of PDEs may lead to increases in cAMP and/or cGMP signalling in undesirable organs, or exacerbated pharmacodynamics in the cardiovascular system. For example, the PATENT PLUS study warned that combining sildenafil with riociguat was associated with systemic hypotension, presumably due to the additive effect on cGMP signalling in the systemic vasculature [170].

In addition, because many PDE enzymes are active in cardiac myocytes [24], PDE inhibition aiming at enhancing cAMP signalling in the lung vasculature may also lead to adverse cardiac effects. These effects potentially emerge from alterations in excitation-contraction coupling in the right and possibly the left ventricles [203]. In line with this notion, PDE3 and PDE4 inhibitions were shown to potentiate spontaneous diastolic Ca^2+^ waves in adult right ventricular myocytes in a model of right ventricular dysfunction in pigs [140]. This may limit the use of bi-selective PDE3-PDE4 inhibitors in PAH, although inhibiting both enzymes may be necessary to provide salutary effects. Refining lung-restricted drug delivery strategies may help overcome this limitation and requires additional studies.

### 4.2. Challenges in Developing Isoform- or Cell-Type-Selective PDE Inhibitors

#### 4.2.1. Identification of Relevant PDE Isoforms and Subcellular Complexes

It is well accepted that PDEs play a key role in regulating the amplitude, duration, and localization of cAMP and cGMP in vascular cells, thereby affecting the extent of vasodilation and PA wall remodelling. Unfortunately, the understanding of their specific contribution to the various cellular processes still remains limited. Therefore, further knowledge is needed as to the distribution of various PDEs in the relevant subcellular compartments. Indeed, the concept of compartmentalization of cyclic nucleotide signalling is becoming increasingly important in the cardiovascular system [24,204]. It is now well accepted that the signalling pathways are spatially organized around different discrete cAMP and cGMP pools within the cell, each regulating specific cellular functions. PDEs are a key component of this organization, together with other proteins (regulatory subunits, kinases, phosphatases, anchoring proteins, etc.). Increases in PDEs expressions under pathological conditions may jeopardize these signalling scaffolds and thereby promote disease progression. Nevertheless, the delineation of cAMP and cGMP compartments in vascular cells is still in its infancy (reviewed in [20,167]) and needs further research efforts. Additionally, data generated from multi-omics will undoubtedly yield important information on PDEs distribution across various cell types. Research based on single-cell approaches should also provide a more accurate identification of the relevant cell subtypes to be targeted.

Therefore, further knowledge in this area will consolidate the definition of new molecular targets involved in PDE signalling.

#### 4.2.2. Toward High-Resolution Targeting?

While various approaches may be considered to selectively target a relevant PDE isoform, these approaches should also address this targeting in the relevant subcellular compartment. Several strategies may be implemented to achieve this ambitious goal, including the development of selective inhibitors of discrete PDE isoforms, disrupting molecular interactions and protein complexes, or various silencing options (extensively reviewed in [5,35]).

Improved organ or even cell-type-selective targeting may benefit from progress made in drug delivery strategies. Nanoformulations administered through the respiratory route may provide restricted delivery of compounds and limit the side effects. Aerosolization of drugs, as experimented with by Schermuly et al. [141], appears as a good starting point for further optimization. For example, efforts to improve the delivery of sildenafil have been proposed [205], which may be extended to other classes of PDE inhibitors. These strategies will face numerous challenges, including tolerability and efficacy compared to standard medications.

## 5. Conclusions

It is well accepted that enhancing cAMP and cGMP signalling in the pulmonary vasculature is beneficial in PAH. PDEs, as natural limitation mechanisms of these pathways, are obvious candidates for therapeutic intervention. Currently, only PDE5i are successfully used in PAH treatment and are still included in the standard battery of first-line therapy. Other PDE families have been the object of characterization in PAH models and tissue from patients. When inhibitors were available, pharmacological therapy was tested with variable efficacy. Research on currently overlooked PDE families, such as PDE10, may benefit from the optimization of pharmacological inhibitors.

Limits in efficacy may arise from redundancy of PDE isoforms, low expression of the targeted PDE, and high severity of the pathological status. In addition, the insufficient selectivity of small molecules may lead to adverse effects due to overly broad PDE inhibition or even off-target effects. Improving understanding of the architecture of the PDE signalling in vascular, cardiac, and inflammatory cells will help define relevant subcellular compartments and enzymes to be targeted. Moreover, the development of nanomedicine and novel drug delivery systems may open up new avenues and accelerate the advancement of novel therapeutic strategies.

Co-targeting of other cyclic nucleotide modulators, such as NP, NO, or G-protein coupled receptors, should be considered to optimize the reshaping of these signalling pathways. Studies focused on possible crosstalk between TGF-β receptor families and cAMP or cGMP will also provide useful translational insights into the understanding of how cyclic nucleotide modulators may interact with sotatercept or future novel therapies.

## Figures and Tables

**Figure 1 cells-14-01670-f001:**
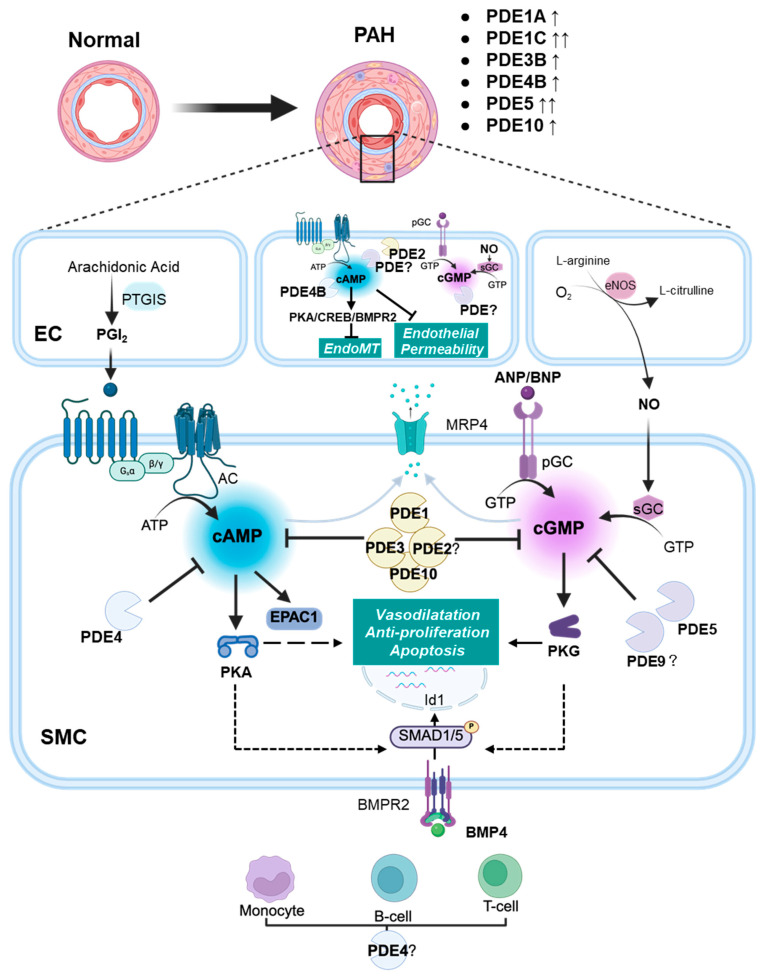
Schematic diagram illustrating the role of PDEs in pulmonary arterial hypertension (PAH). In the pulmonary artery, prostacyclin (PGI_2_) stimulates the production of cAMP, which subsequently activates downstream effectors such as PKA and EPAC1. Cyclic-GMP is generated either by NO-mediated stimulation of sGC or via the stimulation of pGC by natriuretic peptides ANP and BNP. Cyclic-GMP activates PKG. In endothelial cells (ECs), cAMP mitigates endothelial permeability and EndoMT. Within pulmonary artery smooth muscle cells (PASMCs), cyclic nucleotides exert their effects by activating PKG and PKA, targeting multiple downstream pathways involved in vasodilation, cell proliferation inhibition, and apoptosis. The pathway downstream of BMPR2 signalling is promoted by cAMP and cGMP. The reduction in cAMP and cGMP levels occurs either via degradation by PDEs or through export from PASMCs by MRP4. In PAH, upregulation of specific PDEs (e.g., PDE1A, PDE1C, PDE3B, PDE4B, PDE5A, and PDE10A) accelerates the degradation of cAMP and cGMP. This attenuates the beneficial effects of cyclic nucleotides in ECs and PASMCs. In the adventitia, upregulation of PDEs (possibly PDE4) may also contribute to immune cell activation and inflammation, further exacerbating PAH. Solid lines represent direct actions; dashed lines represent indirect actions. AC: adenylyl cyclase; BMPR2: bone morphogenetic protein receptor 2; EC: endothelial cell; EndoMT: endothelial-to-mesenchymal transition; EPAC1: exchange protein activated by cAMP 1; eNOS: endothelial nitric oxide synthase; Id1: inhibitor of DNA binding protein 1; NO: nitric oxide; MRP4: multidrug resistance-associated protein 4; PDE: phosphodiesterase; pGC: particulate guanylyl cyclase; PASMC: pulmonary artery smooth muscle cell; PKA: cAMP-activated protein kinase; PKG: cGMP-activated protein kinase; PGI_2_: prostacyclin_;_ PTGIS: PGI_2_ synthase; sGC: soluble guanylyl cyclase; ↑: modest increase in expression; ↑↑: strong increase in expression.

**Figure 2 cells-14-01670-f002:**
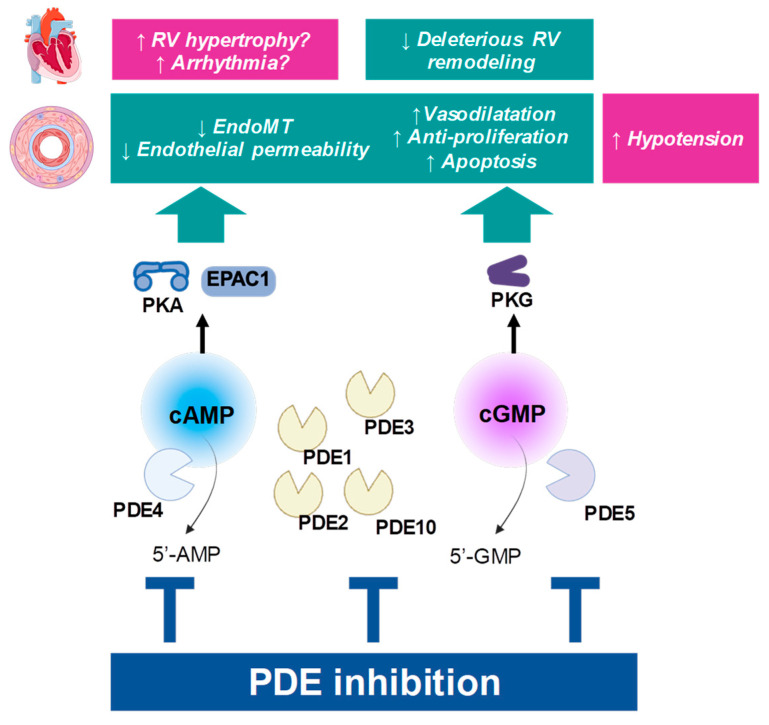
Principle of PDE inhibition. Beneficial influence on PAH pathophysiology is depicted in green boxes, while possible cardiovascular adverse effects are presented in pink boxes. “↑”: the function is increased; “↓”: the function is decreased; RV: right ventricle.

**Table 1 cells-14-01670-t001:** Clinical classification of pulmonary hypertension (PH) [6].

GROUP 1. Pulmonary Arterial Hypertension (PAH)		
	1.1 Idiopathic	
		1.1.1 Not responding to vasoreactivity testing
		1.1.2 Responding to vasoreactivity testing
	1.2 Heritable: mutations of genes: *BMPR2*, *EIF2AK4*, *ACVRL1*, *ENG*…	
	1.3 Associated with drugs and toxins (e.g., fenfluramine, dasatinib)	
	1.4 Associated with:	
		1.4.1 Connective tissue disease
		1.4.2 HIV infection
		1.4.3 Portal hypertension
		1.4.4 Congenital heart disease
		1.4.5 Schistosomiasis
	1.5 PAH with features of venous/capillary (PVOD/PCH) involvement	
	1.6 Persistent PH of the newborn	
GROUP 2. PH associated with left heart disease		
GROUP 3. PH associated with lung diseases and/or hypoxia		
GROUP 4. PH associated with pulmonary artery obstructions		
GROUP 5. PH with unclear and/or multifactorial mechanisms		

PH: pulmonary hypertension, PVOD: PAH with overt features of venous/capillary involvement, HIV: human immunodeficiency virus.

**Table 3 cells-14-01670-t003:** General PDE expression and therapeutic relevance in humans.

PDEs	Main Tissue Expression	Compound (Brand Name, Year of Approval)	Indication and Status
PDE1	brain, smooth muscle, heart, testis	vinpocetine (N.A.)	cerebral vascular disorders and memory impairment; sold as an over-the-counter supplement.
PDE2	adrenal cortex, brain, heart	-	-
PDE3	heart, smooth muscle, adipose tissue, platelets	cilostazol (PLETAL, 1999)	intermittent claudication; second-line therapy
		milrinone (PRIMACOR, 1987)	congestive heart failure; mainly used in surgery and critical care units for hemodynamic support.
		amrinone (INOCOR, 1984)	congestive heart failure; no longer used
		enoximone (N.A.)	congestive heart failure; limited
		anagrelide (AGRYLIN, 1997)	thrombocythaemia; second-line therapy
PDE4	ubiquitous	roflumilast (DALIRESP, 2011)	chronic obstructive pulmonary disease; add-on therapy
		(ZORYVE, 2022)	plaque psoriasis; topical form in dermatology
		apremilast (OTEZLA, 2014)	psoriasis and psoriatic disorders; Behçet’s disease
		crisaborole (EUCRISA, 2016)	moderate atopic dermatitis (patients >2 years old)
		drotaverine (N.A.)	functional bowel disorders: antispasmodics used worldwide
PDE5	smooth muscle, platelets, cerebellum	sildenafil (VIAGRA, 1998)	erectile dysfunction
		(REVATIO, 2005)	PAH
		vardenafil (LEVITRA, 2003) (STAXYN, 2010)	erectile dysfunction
		tadalafil (CIALIS, 2003)	erectile dysfunction, benign prostatic hyperplasia
		(ADCIRCA, 2009)	PAH
		avanafil (STENDRA, 2012)	erectile dysfunction
PDE6	retina	-	-
PDE7	skeletal muscle, immune cells, and brain	-	-
PDE8	immune cells, liver, kidney, testis, thyroid	-	-
PDE9	brain, kidney	-	-
PDE10	brain, testis	papaverine (1938)	visceral and vascular spasm, and erectile dysfunction; not a first-line medication
PDE11	prostate, testis, skeletal muscle		

N.A.: Not approved by the FDA. “-“: none.

**Table 4 cells-14-01670-t004:** Summary of non-clinical explorations on PDEs as therapeutic targets for the treatment of PAH.

PDE	Expression in Patients (RNA Level, Unless Specified)	Effects of Inhibitors on PA Vascular Reactivity	Efficacy of PDE Inhibition in Animal Models					
			Animal Model	Change in Expression	Inhibitor Used In Vivo	Key Findings	Mechanistic Insights	Reference
PDE1	↑ PDE1A (++), RNA, and protein [43,45,52] ↑ PDE1C (+++), RNA, and protein [41,43,52]	8-MM-IBMX: dilates hypoxic rat PA more than normoxic rat PA [43]	rat MCT	↑ PDE1A	8-MM-IBMX	effective (curative protocol)		Schermuly et al., 2007 [43]
			mouse CHx	↑ PDE1A				
			rat cold-induced PAH	↑ PDE1C	8-MM-IBMX	effective (curative protocol)	decrease in macrophage infiltration	Crosswhite and Sun, 2013 [41]
PDE2	↓ [47] ↑ (+) [52]	Bay 60-7750 and EHNA have vasorelaxant effects in rat PA and perfused lung	mouse CHx and bleomycin	↓ PDE2 [52] (hypoxic rat PA)	Bay 60-7550	effective (preventive protocol) effective in combination with other therapies (curative protocol)	increase in GMP and cAMP signalling	Bubb et al., 20l4 [47]
PDE3	↑ PDE3B (+++), RNA, and protein [52] ↑ PDE3A (+) [52]	cilostamide attenuates acute hypoxic vasoconstriction; motapizone dose-dependently relaxes human PA [40]	rat CHx		cilostamide	ineffective alone effective in combination with iloprost or rolipram (preventive)		Phillips et al., 2005 [51]
			rat MCT		cilostazol	effective (preventive and curative protocols)		Chang et al., 2008 [91]; Ito et al., 2021 [92]
			rat CHx		cilostazol	ineffective (preventive)		Ito et al., 2021 [92]
PDE4	PDE4A-D were unchanged in PASMCs [52] ↑ (++) PDE4B in lung, RNA, and protein [93]	rolipram did not relax human PA [40]; rolipram relaxes rat PA [42]	rat CHx		rolipram	ineffective alone effective in combination with iloprost or cilostamide		Phillips et al., 2005 [51]
			rat MCT and CHx		roflumilast	effective (preventive protocols in MCT and CHx rat PA) effective (curative protocol in MCT rat PAH)	reduces interleukin-6 and monocyte chemotactic protein-1	Izikki et al., 2009 [94]
			mouse SuHx	↑ PDE4B in the lung; ↑ PDE4B in PA ECs (under Hx)	roflumilast	effective (preventive protocol)	promotes EndoMT by attenuating the PKA-CREB-BMPR2 axis	Xing et al., 2024 [93]
					PDE4B global KO and EC-specific KO			
			rat SuHx	↑ PDE4B in lung	BI 1015550	effective (curative protocol)		
PDE5	↑ PDE5A, RNA, and protein [52,89]	zaprinast relaxes human PA dose-dependently [40] E4021, sildenafil decreases hypoxic rat PA pressure [50,56]	rat CHx	PDE5 ↑ (IHC)	sildenafil	effective (preventive and curative protocol)		Sebkhi et al., 2013 [50] Baliga et al., 2008 [74]
			rat MCT		sildenafil	effective (curative protocol)	decreases MMP-2 and MMP-9	Schermuly et al., 2004 [95]
			mouse CHx		sildenafil	effective (preventive protocol)		Zhao et al., 2001 [96]; Zhao et al., 2003 [97]
			rat MCT		DDCI	effective (preventive protocol)		Li et al., 2020 [98]
			rat MCT		dihydroquinolin-2(1H)-ones	effective (preventive protocol)		Zhang et al., 2024 [99]
PDE6	↑ PDE “γ” under hypoxia [100]		rat CHx	↑ PDE “γ” under Hx [100]				Not documented
PDE7	↑ (+) [52]							Not documented
PDE8	unchanged [52]							Not documented
PDE9	unchanged [52]		mouse CHx		genetic ablation	ineffective		Kolb et al., 2021 [101]
PDE10	unchanged [52]		rat MCT	PDE10 ↑	papaverine	effective (curative protocol)		Tian et al., 2011 [53]
PDE11	unchanged [52]							not documented

“↑”: increased expression; “↓”: decreased expression; (+): moderate increase; (++): substantial increase (+++): strong increase; CHx: chronic hypoxia (model); EC: endothelial cell; EndoMT: endothelial-to-mesenchymal transition; MCT: monocrotaline (model); PA: pulmonary artery; SuHx: Sugen5416–CHx (model); IHC: immunohistochemistry data.

## Data Availability

Data available on request from the authors. The data that support the findings of this study are available from the corresponding author upon reasonable request. Some data may not be made available because of privacy or ethical restrictions.

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
