# Peer review of "Cyclic Nucleotide Phosphodiesterase Families as Targets to Treat Pulmonary Arterial Hypertension: Beyond PDE5 Inhibitors?"

_cells, 2025, doi:10.3390/cells14211670_

Round 1
Reviewer 1 Report
Comments and Suggestions for Authors
This review provides an extensive and well-integrated synthesis of the current knowledge on the cyclic nucleotide phosphodiesterase (PDE) families in the context of pulmonary arterial hypertension (PAH). This review addresses the molecular biology of PDEs, their expression patterns in the pulmonary and systemic vasculature, and the preclinical and clinical evidence supporting their therapeutic targeting. Additionally, this review explores the potential of PDE inhibitors beyond the well-established PDE5 inhibitors. By combining data from animal models, human tissues, and pharmacological studies, the authors provide researchers and clinicians interested in novel PAH therapies with a comprehensive and detailed resource.
The manuscript is highly impressive and remarkably comprehensive, with a wealth of information and extensive referencing that makes it an invaluable resource, almost encyclopedic in scope. Simultaneously, the richness and density of the content may make it challenging for readers to easily follow the main points throughout. To further enhance readability and accessibility, I would like to offer the following suggestions for the authors’ consideration.
1) Structural reorganization of content
It may be helpful to separate the general background on PAH pathogenesis and therapeutic approaches focusing on non-PDE targets from PDE-specific sections. This could allow readers to first understand the broader context before moving to the detailed discussion of individual PDE families.
2) Addition of an overview figure
Including a schematic diagram that summarizes PAH pathogenesis mechanisms and therapeutic targets, including PDE-related pathways, would provide a clear visual framework.
3) Standardization of PDE-specific descriptions (Section 3. Exploring PDE families in pulmonary arteries and relevance in PAH)
The level of detail and structure varied between the PDE sections. A consistent subsection format, for example, expression pattern, functional role, therapeutic evidence, inhibitors tested, and key findings from animal and human studies, might improve clarity. If feasible, aligning these subsections with the parameters used in the tables could facilitate cross-referencing, although exact matching is not essential.
4) Integration of systemic PDE information
A brief summary of PDE expression and therapeutic relevance in systemic (non-pulmonary) tissues, presented in a manner similar to Tables 2 and 3, may help readers understand the broader pharmacological context of PDEs.
5) Streamlining the conclusion
As future perspectives and limitations are better discussed in a separate section, placing them in a new "Future Directions and Limitations" section could improve the structure. The conclusion could then be condensed into a concise paragraph that highlights the key messages of the review.
Author Response
We thank this reviewer for helpful comments. Please see the following responses to these points.
Comments 1) Structural reorganization of content
It may be helpful to separate the general background on PAH pathogenesis and therapeutic approaches focusing on non-PDE targets from PDE-specific sections. This could allow readers to first understand the broader context before moving to the detailed discussion of individual PDE families.
It may be helpful to separate the general background on PAH pathogenesis and therapeutic approaches focusing on non-PDE targets from PDE-specific sections. This could allow readers to first understand the broader context before moving to the detailed discussion of individual PDE families.
Response 1: this was the actual structure of the manuscript, with general introduction of PAH and therapeutic options in section 1; because PDE5i are a standard therapy for PAH we mention them briefly in this section. From section 2 on, the general signalling involving CN and PDEs are then presented (2.2, 2.3). Changes in cyclic nucleotide stimulation and associated therapies are depicted in section 2.3. The details on PDE families in PAH come in section 3.
For better clarity, we further segmented some parts sections as following (for the first 2 sections).
1. Pulmonary Arterial Hypertension
1.1 Introduction
1.2 Definition, classification, and pathophysiology of PAH
1.3 Current therapeutic options
2. Cyclic nucleotides pathways as therapeutic targets in PAH
2.1 General roles of cyclic nucleotides pathways in circulation
2.2 Molecular determinants of the cAMP and cGMP pathway
2.2.1 Synthesis of cGMP and cAMP
2.2.1.1 cGMP
2.2.1.2 cAMP
2.2.2 Hydrolysis of cGMP and cAMP by PDEs
2.3 General modulation of cAMP and cGMP pathways in PAH
2.3.1 Alterations of cAMP and cGMP levels in PAH
2.3.2 Therapies to stimulate the cGMP pathway in PAH
2.3.2.1 Promoting the NO–cGC axis
2.3.2.2 Promoting the NP system
2.3.3 Therapies to stimulate the cAMP pathway in PAH
Comments 2) Addition of an overview figure
Including a schematic diagram that summarizes PAH pathogenesis mechanisms and therapeutic targets, including PDE-related pathways, would provide a clear visual framework.
Response 2: We inserted Figure 1, which recapitulates the main cAMP and cGMP pathways and PDE influence in PAH pathogenesis.
Comments 3) Standardization of PDE-specific descriptions (Section 3. Exploring PDE families in pulmonary arteries and relevance in PAH)
The level of detail and structure varied between the PDE sections. A consistent subsection format, for example, expression pattern, functional role, therapeutic evidence, inhibitors tested, and key findings from animal and human studies, might improve clarity. If feasible, aligning these subsections with the parameters used in the tables could facilitate cross-referencing, although exact matching is not essential.
Response 3: for the subsections related to each PDE family in part 3, “Exploring PDE families in pulmonary arteries and relevance in PAH”, the following standardized structure has been used:
1 Enzymatic properties
2 Expression pattern
3 Functional role and therapeutic potential
4 Perspectives and limitations
These subsections correspond to some items presented in Table 2 or Table 4.
Comments 4) Integration of systemic PDE information
A brief summary of PDE expression and therapeutic relevance in systemic (non-pulmonary) tissues, presented in a manner similar to Tables 2 and 3, may help readers understand the broader pharmacological context of PDEs.
Response 4: A new Table was inserted (Table 3): “General PDE expression and therapeutic relevance in human.”
Comments 5) Streamlining the conclusion
As future perspectives and limitations are better discussed in a separate section, placing them in a new "Future Directions and Limitations" section could improve the structure. The conclusion could then be condensed into a concise paragraph that highlights the key messages of the review.
Response 5: As recommended, a new section was inserted:
- Future directions and limitations
4.1 Potential systemic side effects of PDE inhibition (e.g., hypotension, cardiac effects).
4.2 Challenges in developing isoform- or cell-type-selective PDE inhibitors
4.2.1 Identification of relevant PDE isoforms and subcellular complexes
4.2.2 Toward high-resolution targeting?
The “5. Conclusion” section was rewritten to summarize key facts and perspectives.
Reviewer 2 Report
Comments and Suggestions for Authors
This comprehensive review provides a timely and detailed overview of the role of phosphodiesterase (PDE) families in pulmonary arterial hypertension (PAH), with a particular focus on therapeutic potential beyond PDE5 inhibition. The authors effectively provide a large body of preclinical and clinical literature, highlighting both established and emerging targets. The manuscript is well-structured, clearly written, and addresses a topic of significant clinical and pharmacological interest. The inclusion of detailed tables summarizing PDE expression and inhibitor effects is particularly valuable. The review is generally balanced and informative, though several areas could be strengthened to enhance its impact and clarity.
- Introduction: consider briefly mentioning the recent approval of sotatercept earlier in the introduction to frame the evolving therapeutic landscape.
- The review would benefit greatly from a schematic figure illustrating. Such a figure would significantly enhance readability and help readers visualize the complex interactions described. I have privided some ideas below;
- The compartmentalization of cAMP/cGMP signaling in pulmonary vascular cells.
- The specific roles of different PDE families in regulating these pathways.
- How PDE upregulation in PAH contributes to disease pathogenesis and how inhibition may confer benefit.
- The review would be improved by a dedicated subsection or paragraph discussing;
- Challenges in developing isoform- or cell-type-specific PDE inhibitors.
- Potential systemic side effects of PDE inhibition (e.g., hypotension, cardiac effects).
- Table 3 is informative but could be more user-friendly. Consider;
- Standardizing symbols (e.g., ↑, ↓) and ensuring consistency.
- Adding a column for "Key Findings" or "Mechanistic Insights" to summarize the most important results.
- Clearly distinguishing between preventive and curative effects in animal models.
- Some references are outdated.
- The conclusion could be more forward-looking. Suggest specific research priorities (e.g., developing lung-specific delivery systems, leveraging single-cell omics to map PDE expression, exploring PDE interactions with novel PAH targets).
Author Response
We would like to thank Reviewer 2 for helpful comments. Please see the point by point responses below.
Comment 1: Introduction: consider briefly mentioning the recent approval of sotatercept earlier in the introduction to frame the evolving therapeutic landscape.
Response 1: Mention of sotatercept was inserted in the 1.1 Introduction section (lines 37-41):
“Recently, sotatercept, a first-in-class activin signalling inhibitor, has yielded promising results in clinical trials [2,3]. Sotatercept can now be combined with classical PAH therapies [4], and future clinical investigations will aim at refining the strategies for optimal efficacy and safety."
2: typo, no response applicable
Comment 3: The review would benefit greatly from a schematic figure illustrating. Such a figure would significantly enhance readability and help readers visualize the complex interactions described. I have privided some ideas below;
- The compartmentalization of cAMP/cGMP signaling in pulmonary vascular cells.
- The specific roles of different PDE families in regulating these pathways.
- How PDE upregulation in PAH contributes to disease pathogenesis and how inhibition may confer benefit.
Response 3: We inserted Figure 1, which recapitulates the main cAMP and cGMP pathways and PDE influence in PAH pathogenesis. Compartmentalization of CNs and contributions of PDEs was discussed throughout the manuscript. As mentioned, little data exist on compartments in pulmonary vascular cells. We refer to other reviews specifically focused on this subject:
(ref168) Lorigo, M.; Oliveira, N.; Cairrao, E. PDE-Mediated Cyclic Nucleotide Compartmentation in Vascular Smooth Muscle Cells: From Basic to a Clinical Perspective. J Cardiovasc Dev Dis 2021, 9, doi:10.3390/jcdd9010004.
(ref20) Vina, D.; Seoane, N.; Vasquez, E.C.; Campos-Toimil, M. cAMP Compartmentalization in Cerebrovascular Endothelial Cells: New Therapeutic Opportunities in Alzheimer's Disease. Cells 2021, 10, doi:10.3390/cells10081951.
Some data were obtained using cultured aortic smooth muscle cells and cannot be directly extrapolated to the pulmonary artery. An extensive description of the roles of PDEs and the benefits of PDE inhibition is recapitulated in Table 4. Additionally, we added Figure 2, which introduces the principles of PDE inhibition and expected beneficial and adverse cardiovascular effects.
Comment 4: The review would be improved by a dedicated subsection or paragraph discussing;
- Challenges in developing isoform- or cell-type-specific PDE inhibitors.
- Potential systemic side effects of PDE inhibition (e.g., hypotension, cardiac effects).
Response 4: As recommended, such sections were inserted as part of new sections:
- Future directions and limitations
4.1 Potential systemic side effects of PDE inhibition (e.g., hypotension, cardiac effects)
4.2 Challenges in developing isoform- or cell-type-selective PDE inhibitors
4.2.1 Identification of relevant PDE isoforms and subcellular complexes
4.2.2 Toward high-resolution targeting?
Comment 5: Table 3 is informative but could be more user-friendly. Consider;
- Standardizing symbols (e.g., ↑, ↓) and ensuring consistency.
- Adding a column for "Key Findings" or "Mechanistic Insights" to summarize the most important results.
- Clearly distinguishing between preventive and curative effects in animal models.
Response 5: Table format has been checked for consistency and corrected. Headings of the columns were changed as proposed. We thought that summarizing the sublines of the columns would detract from the table, which already provides a quick overview of the key outcomes of each study. When relevant, this distinction between preventive or curative protocols was specified in the table and also in the whole manuscript
e.g.:
l444: “Nevertheless, the PDE2 inhibitor alone, administered from day 14, once PH was established, was not effective in curing PH.”;
l353: “Pharmacological treatments were administered for 1 to 2 weeks once the disease was established, following a curative-like strategy."
Comments 6: Some references are outdated.
Response 6: The relevance of old references has been checked. Where necessary, outdated references were updated.
Some old references were left, with the following justifications:
ref8. D'Alonzo et al. 1991: provides survival rates for the period before the area of PAH-specific therapy. (L78). This is relevant as this paragraph depicts the evolution of therapies through the last decades.
MacLean et al., 1996 (ref54) and 1997 (ref39); Rabe et al., 1994 (ref40); Haynes et al., 1996 (ref46); Suttorp et al., 1993 (48) and 1996 (ref 46), Cohen et al., 1996 (ref56): these studies were pioneer in exploring expression and activity of PDEs in pathological pulmonary circulation.
Pepke-Zaba et al., 1991 (ref63); Giaid and Saleh, 1995 (ref60), Roberts et al., 1993 (ref65), Beghetti et al., 1995 (ref66), Adatia et al., 1995 (ref78), Christman et al., 1992 (ref78) and 1993 (ref79), Higenbottam et al., 1984 (ref82); Barst et al., 1996 (ref 83): these classic studies report important exploration and therapeutic efforts in PAH patients.
Ahn et al., 1989 (ref108); Wells et al., 1988 (ref109), Mery et al, 1995 (ref124): report first characterizations of classic PDE inhibitors.
Others are princeps papers presenting the discovery and cloning of PDE genes.
All these articles provide valuable information for those interested in obtaining further details on PDE biochemistry, pharmacology, and physiology.
Comments 7: The conclusion could be more forward-looking. Suggest specific research priorities (e.g., developing lung-specific delivery systems, leveraging single-cell omics to map PDE expression, exploring PDE interactions with novel PAH targets).
Response 7: Suggestions were added in the conclusion and in a new section 4: “4. Future directions and limitations”: e.g.,
l848: “Refining lung-restricted drug delivery strategies may help overcome this limitation and requires additional studies.”
l864: "Nevertheless, the delineation of cAMP and cGMP compartments in vascular cells is still in its infancy (reviewed in [20,168]) and needs further research efforts. Additionally, data generated from multi-omics will undoubtedly yield important information on PDEs distribution across various cell types. Research based on single-cell approaches should also provide a more accurate identification of the relevant cell subtypes to be targeted.
Therefore, further knowledge in this area will consolidate the definition of new molecular targets involved in PDE signalling.”
l873: “While various approaches may be considered to selectively target a relevant PDE isoform, these approaches should also address this targeting in the relevant subcellular compartment.”
l894: ”Research on currently overlooked PDE families, such as PDE10, may benefit from the optimization of pharmacological inhibitors.”
l899: “Improving understanding of the architecture of the PDE signalling in vascular, cardiac, and inflammatory cells will help improve the definition of relevant subcellular compartments and enzymes to be targeted. Moreover, the development of nanomedicine and novel drug delivery systems may open new avenues and accelerate novel therapeutic strategies.”
l906: “Studies focused on possible crosstalk between TGF-beta receptor families and cAMP or cGMP will also provide useful translational insight, as to the understanding of how cyclic nucleotides modulators may interact with sotatercept or future novel therapies."
Reviewer 3 Report
Comments and Suggestions for Authors
The manuscript presents a comprehensive review of phosphodiesterase (PDE) families as potential therapeutic targets in pulmonary arterial hypertension (PAH). The topic is important, as current treatments are limited and PDE5 inhibitors dominate clinical practice.
The review could be valuable, but in its current form it is overly descriptive, lacks critical synthesis, and is difficult to read.
A restructuring and sharper focus on clinical translation are needed:
-While the manuscript summarizes a large amount of background data, it often reads as a compilation of studies rather than a critical review. The authors should clearly highlight the strengths/weaknesses of the available data and identify knowledge gaps. For example, sections on PDE1, PDE2, PDE3, and PDE4 list expression data and inhibitor effects but rarely evaluate the translational value or limitations.
-Large parts of the text reproduce mechanistic details that are well-established in textbooks and prior reviews (for example the general cyclic nucleotide pathways). The authors should focus on novel findings and therapeutic implications. Repetition of basic information makes the manuscript too long and less impactful.
-Although the title promises discussion "beyond PDE5 inhibitors," clinical perspectives on alternative PDE inhibitors are not sufficiently developed. Most evidence presented is preclinical, with little effort to link findings to feasibility of clinical translation. Authors should discuss why other PDE families have failed to advance clinically, and what barriers persist.
-The manuscript is dense, with excessive detail in tables and long subsections. A more structured approach —summarizing each PDE family with a concise "Key evidence – Limitations – Therapeutic potential" format— would improve clarity.
-Table 2 is useful but overwhelming in detail. Condensing it to highlight the most therapeutically relevant PDE families would help. Well-designed visuals would also help.
Comments on the Quality of English LanguageGrammatical and stylistic errors (e.g., "rich of 11 families" should be "comprising 11 families"; "therapeutical" should be "therapeutic"). A thorough language edit is required.
Author Response
We would like to thank this reviewer for these helpful comments. Please see our point by point responses below .
Comment 1) A restructuring and sharper focus on clinical translation are needed:
-While the manuscript summarizes a large amount of background data, it often reads as a compilation of studies rather than a critical review. The authors should clearly highlight the strengths/weaknesses of the available data and identify knowledge gaps. For example, sections on PDE1, PDE2, PDE3, and PDE4 list expression data and inhibitor effects but rarely evaluate the translational value or limitations.
Response 1: Data reporting has been checked and rewritten in a more critical and synthetic manner. References to other systems and studies have been added. For each PDE family, a new “Perspective and limitations” section identifies knowledge gaps and proposes ways to improve knowledge and address the translational issues. Additionally, Section #4 has been added to further discuss these points.
Some examples of synthetic, critical reviewing, as requested, are listed below:
l341: "This is consistent with the role played by PDE1C in promoting vascular SMC proliferation in other diseases, where vessel wall remodeling was involved [5,103,105,106]."
l362: “Although vinpocetine [108] and 8-MM-IBMX [109][106] are commonly used PDE1 inhibitors, the selectivity of these compounds for PDE1 is moderate. (…) Therefore, pharmacological effects should be interpreted cautiously. However, some promising PDE1 selective inhibitors have emerged. (…). As previously discussed, because expression patterns in animal models may not accurately mimic those in patients, the translational value of in vivo studies remains uncertain. Data from human tissue models or clinical studies are clearly needed to ascertain the therapeutic potential of recent PDE1 inhibitors in PAH.”
l379: "Considering that PDE1A and C are expressed in the RV [116] and that their expression rises in the failing ventricle [5], PDE1 inhibition may promote cardiac effects in the hypertrophied RV of PAH patients. Whether these cardiac and vascular effects would translate into beneficial (improved RV function) or deleterious (arrhythmias) outcomes remains uncertain and needs to be investigated."
l463: "Considering the various roles that PDE2 is thought to play in endothelial cells (promoting angiogenesis and barrier leakage) and in the hypertrophied cardiomyocyte (reviewed in [118]), the efficacy and safety of PDE2 inhibition in pulmonary circulation require further exploration using different models. (...) Furthermore, a combination with other PDE inhibitors or therapies that enhance cyclic nucleotides would be more likely to reach efficacy endpoints."
l517: “Consistent with ubiquitous PDE3 expression through smooth muscle cells [27], the vasodilatory effects of PDE3 inhibitors, or other mixed and non-selective inhibitors, extend to systemic vasculature and promote hypotension. This is a limitation to straightforward translation of these compounds for use in clinics [133,136]."
l612: "Nevertheless, further investigations are necessary to explore the long-term potential of this therapy. One may speculate that similar limitations may apply to PDE4 inhibitors as to PDE3 inhibitors, although the level of expression of PDE4 is much lower than PDE3 in human cardiac tissue, thereby limiting the potential inotropic and proarrhythmic effects of a possible PDE4 inhibition-based therapy [24]."
l627: "The clinical development of PDE4 inhibition has been hindered by adverse side effects, mainly in the gastrointestinal area [5]. (...) The use of isoform-selective inhibitors, such as the anti-fibrotic nerandomilast (BI 101555), may offer therapeutic insight through future dedicated clinical investigation."
l696: "The role of PDE5 as a master controller of cGMP limits the use of further combinational therapy with other cGMP enhancers in clinics. (...) Recently, novel highly selective PDE5i have been reported (...) These new compounds may offer improved selectivity compared to sildenafil or tadalafil, improving the tolerability of PDEi among patients.
l810: "As discussed previously, the selectivity of papaverine for PDE10 is limited. Therefore, efforts using highly selective inhibitors of PDE10 may help demonstrate the translational potential in PH. Several PDE10 inhibitors have been developed in the central nervous system area (...) Such growing diversity of selective PDE10 inhibitors may afford opportunities for potent pharmacological approaches to treat PAH. Nevertheless, because PDE10 is abundant in the brain, strategies that favor pulmonary and cardiac selectivity would be valuable to preserve the tolerability of future drug candidates."
Comment 2) -Large parts of the text reproduce mechanistic details that are well-established in textbooks and prior reviews (for example the general cyclic nucleotide pathways). The authors should focus on novel findings and therapeutic implications. Repetition of basic information makes the manuscript too long and less impactful.
Response 2: Section 2 has been shortened, and Figures 1 and 2 present general cyclic nucleotide pathways. The whole manuscript has been reviewed and rewritten in a more synthetic and impactful manner. Mechanisms specific to PAH have been added, e.g.,
l119: “Enhancing both cAMP and cGMP pathways results in promoting BMPR2 and Smad1/5 signallings, which are important for maintenance of the pulmonary vascular integrity [22,23]. Because sotatercept also aims to restore these pivotal mechanisms, cyclic nucleotide-based therapies may provide supplemental benefit in targeting the TGF-b axis."
Comment 3) -Although the title promises discussion "beyond PDE5 inhibitors," clinical perspectives on alternative PDE inhibitors are not sufficiently developed. Most evidence presented is preclinical, with little effort to link findings to feasibility of clinical translation. Authors should discuss why other PDE families have failed to advance clinically, and what barriers persist.
Response 3: As depicted in the review, few PDE-inhibiting compounds have been clinically tested. When possible, updates of current clinical trials or emerging pharmaceutical were mentioned. Frequent discussions on limitations due to adverse effects or low efficacy have been included.
e.g. , please see sections "perspectives and limitations":
- section 3.4.4, starting l516,
- section 3.5.4, starting l607
Comment 4) -The manuscript is dense, with excessive detail in tables and long subsections. A more structured approach —summarizing each PDE family with a concise "Key evidence – Limitations – Therapeutic potential" format— would improve clarity.
Response 4: for the subsections related to each PDE family in part 3, “Exploring PDE families in pulmonary arteries and relevance in PAH”, the following standardized structure has been used:
- Enzymatic properties
- Expression pattern
- Functional role and therapeutic potential
- Perspectives and limitations
The contents on “enzymatic properties” and “expression pattern” have been lightened. Still we would like to leave these sections as they provide important information for those would like to understand the mechanistic role of PDEs and the consequences of their inhibition. Moreover, data on expression can be complex (smooth muscle vs. endothelial) and controversial.
Comment 5) -Table 2 is useful but overwhelming in detail. Condensing it to highlight the most therapeutically relevant PDE families would help. Well-designed visuals would also help.
Response 5: The format of the Tables has been reviewed. It is unclear whether this comment refers to former Table 2 or Table 3. Our intention is to provide the reader with a comprehensive overview of all the PDE families, whether their therapeutic relevance is strong or only potential.
Comment 6) Comments on the Quality of English Language
Grammatical and stylistic errors (e.g., "rich of 11 families" should be "comprising 11 families"; "therapeutical" should be "therapeutic"). A thorough language edit is required.
Response 6: These corrections have been done. The quality of the English language has been extensively reviewed with the assistance of a staff member of the Academic Writing Center at the Université Paris-Saclay. We hope this has substantially improved the quality of English.
Round 2
Reviewer 2 Report
Comments and Suggestions for Authors
The authors have addressed all suggestions. No further comments are required.
Reviewer 3 Report
Comments and Suggestions for Authors
the authors improved a lot the manuscript.